# Learning in a sensory cortical microstimulation task is associated with elevated representational stability

Ravi Pancholi[1], Lauren Ryan [1] & Simon Peron [1] ✉

Sensory cortical representations can be highly dynamic, raising the question of how representational stability impacts learning. We train mice to discriminate the number of photostimulation pulses delivered to opsin-expressing pyramidal neurons in layer 2/3 of primary vibrissal somatosensory cortex. We simultaneously track evoked neural activity across learning using volumetric two-photon calcium imaging. In well-trained animals, trial-to-trial fluctuations in the amount of photostimulus-evoked activity predicted animal choice. Population activity levels declined rapidly across training, with the most active neurons showing the largest declines in responsiveness. Mice learned at varied rates, with some failing to learn the task in the time provided. The photoresponsive population showed greater instability both within and across behavioral sessions among animals that failed to learn. Animals that failed to learn also exhibited a faster deterioration in stimulus decoding. Thus, greater stability in the stimulus response is associated with learning in a sensory cortical microstimulation task.

Cortical activity can be highly dynamic, even in the context of identical sensory input. In olfactory[1], visual[2–5], somatosensory[6,7], auditory[8], multimodal[9], and motor cortical areas[10,11], as well as hippocampus[12], the population of neurons representing a particular sensory or motor feature changes over time. This turnover in the stimulus-responsive population is known as representational drift[13,14]. Because variance in neural activity within sensory cortices can disrupt readout in downstream areas, representational drift is viewed as a potential constraint on stable perception and behavior[15]. At the same time, its ubiquity in both neocortex and allocortex suggests that it is a fundamental cortical feature.

Representational stability is typically studied in animals receiving input at the sensory epithelium. Under such conditions, dynamics in the sensory stream prior to cortex can contribute to variability in the cortical response. Moreover, fanout at pre-cortical processing stages distributes information across multiple cortical and subcortical targets, reducing the relative perceptual importance of the cortical area under study[16]. Microstimulation of cortex circumvents pre-cortical processing and simplifies interpretation of behavioral effects by explicitly requiring the animal to use specific features of the evoked activity. Microstimulation has been used extensively in the study of perception[17–23], and animals can be trained to report microstimulation[24–27]. However, electrical microstimulation of cortex evokes activity among a sparse and spatially distributed population[28], suffers from instability due to slowly emerging gliosis[29], and produces a prominent stimulus artifact that complicates simultaneous recording[30]. Optical microstimulation of cortex[23,26,27,31–33] overcomes these constraints: opsin expression can be stable for the duration of the experiment, viral opsin delivery allows for refined spatial and genetic control of evoked activity, and optical microstimulation is compatible with chronic concurrent recording via large-scale calcium imaging[33,34]. Consequently, optical microstimulation of cortex in conjunction with population imaging is well suited to the study of representational stability and its impact on behavior.

Here, we train mice to report the intensity of optical microstimulation in a subset of opsin-expressing pyramidal neurons in layer (L) 2/3 of primary vibrissal somatosensory cortex (vS1). Mice must discriminate between a high and low number of optical microstimulation pulses delivered to vS1. Over the course of training, we track neural dynamics in the opsin-expressing tissue using volumetric

[1]Center for Neural Science, New York University, 4 Washington Place Rm. 621, New York, NY 10003, USA. ✉e-mail: speron@nyu.edu

two-photon calcium imaging[34]. Photostimulus-evoked activity declines rapidly over the course of learning, especially among the most responsive neurons. Photostimulus-evoked activity also predicts animal choice in well-trained mice. Mice exhibit different rates of learning. Throughout learning, evoked activity in the photoresponsive population changes at rates that vary across animals. Animals with more stable neural activity, both within and across behavioral sessions, are more likely to learn the task. These animals exhibit a slower decline in the efficacy of stimulus decoding using neural activity. Together, our results show that mice that learn the task exhibit greater stability of evoked activity in sensory cortex.

## Results

### Mice can learn an optical microstimulation pulse count discrimination task

We monitored cortical dynamics as mice were trained on an optical microstimulation task. Transgenic mice expressing GCaMP6s[35] in

cortical excitatory neurons (Slc17a7-Cre X Ai162)[36] were virally transfected with the soma-restricted opsin ChRmine (AAV-8-CaMKIIa-ChRmine-mScarlet-Kv2.1-WPRE)[37], resulting in opsin expression in pyramidal neurons in L2/3 of vS1. To allow for optical access, a cranial window was implanted over vS1 (Fig. 1a). Following recovery, we used widefield imaging during whisker deflection to confirm that opsin expression was restricted to vS1 (typically the C row of whiskers; Supplementary Fig. 1). A miniature light-emitting diode (LED) was affixed to the cranial window adjacent to the site of opsin expression for optogenetic stimulation.

Light pulses (5 ms long; minimum inter-pulse onset interval, 50 ms) were delivered to the opsin-expressing area and animals were required to report whether the pulse count was high or low. Mice were rewarded for licking the left of two lickports on low pulse count trials (1 or 3 pulses; 0 pulses in early training) and the right lickport on high pulse count trials (7 or 9 pulses), with 5 pulse count trials rewarded randomly between left and right (Fig. 1b). On each trial, photostimulus

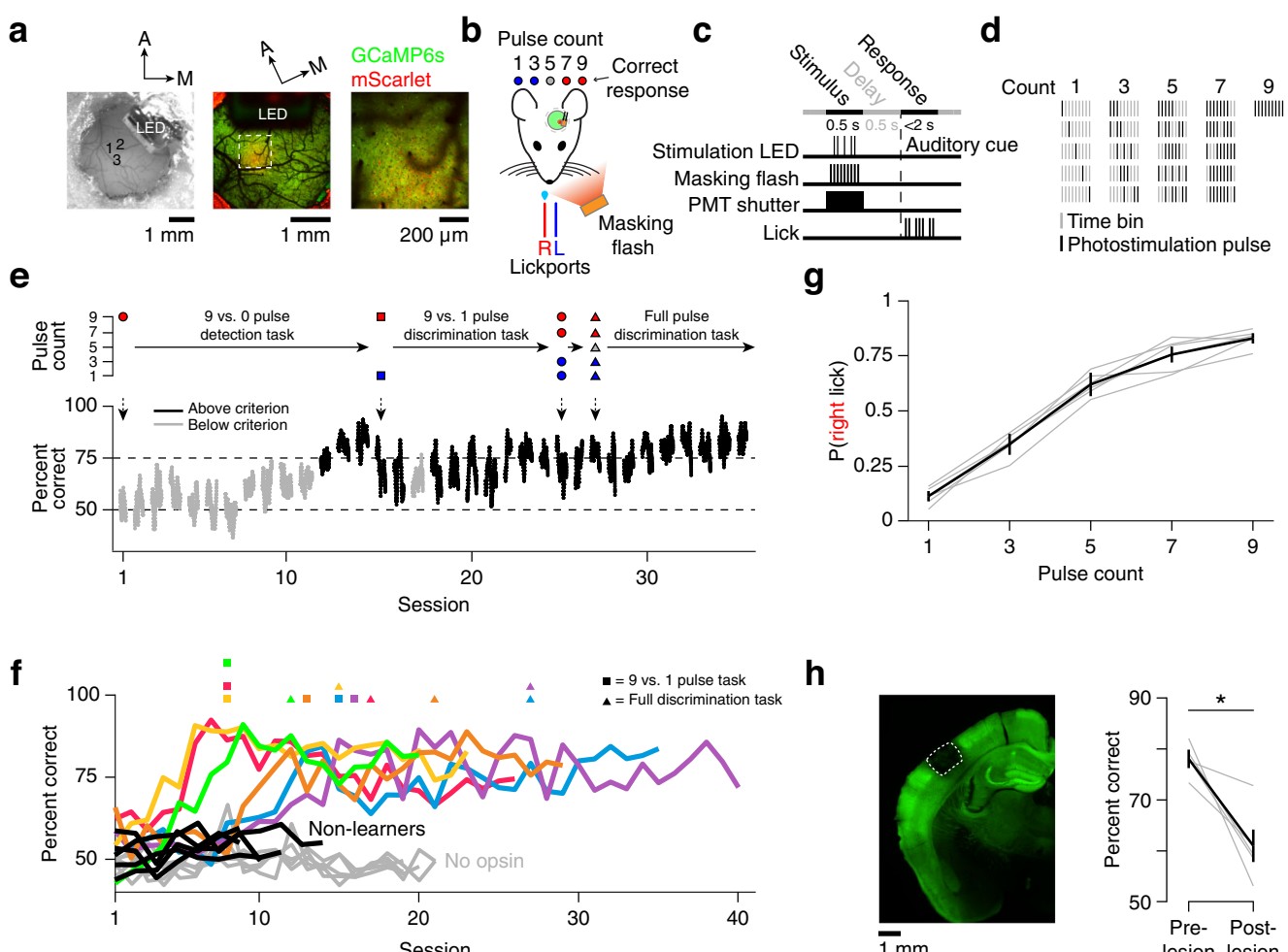

**Fig. 1 | Mice learn to discriminate photostimulus pulse count using evoked vS1 activity. a** Opsin expression in barrel cortex. Left, widefield view of cranial window showing viral injection sites and LED; center, two-photon image of cranial window (green, GCaMP6s fluorescence; red, mScarlet fluorescence); right, higher magnification two-photon image. **b** Mice must lick the right (red) lickport following presentation of 7 or 9 LED pulses or the left (blue) lickport following presentation of 0, 1, or 3 pulses to obtain reward. 5 pulses are randomly rewarded. A spectrally matched masking flash LED was used throughout. **c** Task timing during a single trial (Methods). **d** Example photostimulation pulse timing permutations. **e** Performance during learning. Top, pulse count progression; bottom, task performance as a function of training day for one animal, with each dot indicating the rolling average

of 61 trials centered on the shown trial (Black, days where peak d-prime ≥ 1.5; grey, d-prime < 1.5). **f** Task performance as a function of training day for individual animals. Colored, animals expressing opsin that learned the task; black, animals expressing opsin that failed to learn the task; grey, animals with no opsin. **g** Right lick probability as a function of pulse count in final version of the task. Grey, individual animals; black, mean ± standard error of the mean (s.e.m.), $n = 6$ mice. **h** Lesion of opsin-expressing tissue. Left, coronal section following laser lesion (box, lesion extent; green, GCaMP6s fluorescence); right, task performance before and after lesion. Grey, average performance over 3 sessions pre- and post-lesion for individual animals; black: mean ± s.e.m. *$P < 0.05$ for paired two-sided *t*-test ($P = 0.014$).

pulses were presented during the stimulus epoch (500 ms). This was followed by a short delay period (500 ms), after which mice indicated their response by licking during the response epoch (<2 s) (Fig. 1c). LED output was stable within and across training sessions (Supplementary Fig. 2). Increasing the pulse count, the pulse amplitude, or the inter-pulse interval all resulted in greater photostimulation-evoked activity in awake animals not performing a task[31] (Supplementary Fig. 3). Stimulus pulses were randomly permuted among 9 time bins to prevent animals from solving the task using timing cues (Fig. 1d, Supplementary Fig. 4). A spectrally matched LED was placed near the mouse's eyes and illuminated during each time bin to obscure potential visual cues from photostimulation ('masking flash', Fig. 1b).

Mice (Supplementary Table 1) were first trained on a photostimulation detection task (9 vs. 0 pulses; Fig. 1e). Mice were classified as 'learners' ($n = 6$) if their performance met our learning criterion (at least two consecutive days with a peak d-prime of 1.5 in a 61 trial window; Methods) within the first 2 weeks of training. Mice that failed to improve on this detection task were classified as 'non-learners' ($n = 5$). These mice were only exposed to the 9-pulse vs. 0-pulse stage of the task and were removed from the training cohort within 1-2 weeks. Learners reached criterion performance in $9.8 \pm 3.7$ days (mean ± SD), after which additional pulse counts were added until mice were proficient at the full intensity discrimination task (Fig. 1f, g). Learning was marked by increased performance on both trial types, with decrements in performance on task stage transitions observed only in some animals (Supplementary Fig. 5; Fig. 1e). Learners achieved high performance on the final stage of the task (78.3% ± 3.4% correct).

To confirm that animals required opsin expression to perform the task, we trained a separate cohort of mice not infected with opsin ('no opsin'; Fig. 1f). These mice could not learn the task after 20 training sessions (performance on final day: 48.6% ± 1.4% correct; $n = 5$ mice). Lesioning the opsin-expressing area in mice that had progressed to the final task degraded performance (average performance of 3 sessions before lesion: 78.8% correct, 3 sessions after lesion: 60.9%, $P = 0.014$, paired two-sided $t$-test; Fig. 1h), with performance sometimes remaining above chance because the lesions did not remove all opsin-expressing neurons (Supplementary Fig. 6). Thus, mice can learn to discriminate the number of photostimulus pulses delivered to a subset of neurons in L2/3 of vS1 via optical microstimulation.

## Optical microstimulation response increases with pulse count

To assess neural activity during optical microstimulation, we recorded from L2/3 of the opsin-expressing area using volumetric two-photon calcium imaging[6,34]. Neurons distributed among three 800-by-800 μm planes spaced 60 μm apart were tracked for $29.8 \pm 6.9$ days (mean ± SD; $n = 6$ mice; Fig. 2a). Imaged neurons were separated into opsin-expressing ($692.6 \pm 123.2$ neurons per mouse) and opsin non-expressing ($2287.3 \pm 466.9$ neurons per mouse) populations based on the presence of a co-expressed fluorophore, mScarlet (Methods). Based on estimates of L2/3 pyramidal neuron density in vS1[6,38], we imaged 15–20% of infected L2/3 neurons, for a total of ~3500 to ~4500 opsin-expressing neurons per mouse. In well-trained mice performing the final stage of the task (Fig. 1e, f), neurons exhibited diverse responses to photostimulation (Fig. 2b, c). We found both opsin-expressing and opsin non-expressing excitatory neurons that responded to photostimulation, with photoresponsive neurons showing stronger responses with greater pulse counts (Fig. 2d). Opsin non-expressing neurons were presumably driven by opsin-expressing neurons in a feedforward manner (Fig. 2b). Photoresponsive neurons were broadly spatially distributed (Fig. 2e). Higher photostimulus pulse count consistently increased the fraction of neurons responsive to stimulation (Fig. 2f), as well as the response amplitude (Fig. 2g) and response probability (Fig. 2h) of individual neurons. This was true for both opsin-expressing and opsin non-expressing neurons. In all cases, the opsin-expressing population responded more reliably and strongly to photostimulation than the opsin non-expressing population.

## The amount of microstimulation-evoked activity predicts animal choice

Higher microstimulation pulse counts drive larger responses in photoresponsive neurons. Thus, mice could solve the task by licking in one direction if activity fell below a threshold and licking in the other if activity exceeded a threshold. If mice employed this strategy, variability in evoked activity for a given pulse count should influence choice. Specifically, for a given pulse count, trials where the animal reported more stimulation (by licking right) should show more evoked activity than trials where the animal reported less stimulation (by licking left). In well-trained mice performing the final stage of the task, we found that a subset of individual neurons indeed showed this pattern across all pulse counts, whereas other neurons exhibited a difference on only a subset of pulse counts or no difference at all (Fig. 3a, b).

Sensory decisions are believed to result from pooling across many neurons[39,40]. Therefore, we next asked whether the predicted difference in evoked activity was also present at the scale of the photoresponsive population rather than just among individual neurons. Among strongly photoresponsive opsin-expressing neurons (response probability > 0.25) during an example session, we found that the mean evoked ΔF/F across trials was no different for left and right lick trials (Fig. 3c). Among strongly photoresponsive opsin non-expressing neurons, however, the population response was higher on trials where the animal indicated a stronger stimulus for all pulse counts. Across mice, the peak-normalized response on lick-left and lick-right trials did not differ across pulse counts for opsin-expressing neurons (Fig. 3d, e; $P = 0.060$, two-sided $t$-test comparing right and left lick trials, $n = 6$ mice). For opsin non-expressing neurons, the difference was significant at all pulse counts ($P = 0.017$), with greater levels of evoked activity on trials where the animal reported higher activity. Thus, the amount of evoked activity in a subset of opsin non-expressing neurons downstream from the opsin-expressing population can predict animal choice.

## Activity levels decline over time, but overall activity is not predictive of learning

Having observed a link between neural activity and choice in well-trained animals performing our task, we next examined the evolution of cortical activity as mice became proficient at the task. Sensory cortical representations in L2/3 exhibit diverse dynamics during learning, with changes in aggregate responsiveness as well as in the responses of single neurons[41]. We first asked whether aggregate responsiveness changed across training (Fig. 4a, b). Restricting our analysis to trials with 9 photostimulus pulses, which were present in all sessions, we identified both opsin-expressing and opsin non-expressing neurons with stable, increasing, or decreasing responses to photostimulation (Fig. 4c–e). Aggregate responsiveness did not change significantly in either the opsin-expressing or opsin non-expressing population, though neurons with the highest responsiveness across sessions (neurons with an overall responsiveness in excess of the top 5% or 1% observed across all sessions; Methods) did show declines in responsiveness (Fig. 4f). Despite this decline in responsiveness, high pulse counts evoked more activity among both opsin-expressing and opsin non-expressing neurons than low pulse counts at all stages of training (Supplementary Fig. 7). Thus, training results in a sparsification of the photostimulation response, primarily due to declining responses among the most responsive neurons.

To control for changes in responsiveness due to changes in opsin expression, we tracked mScarlet fluorescence ('redness') over the course of training in mice that learned the task (Supplementary Fig. 8; Methods). We found that overall redness across all neurons did not change over time ($P = 0.490$, paired two-sided $t$-test, $n = 6$ mice;

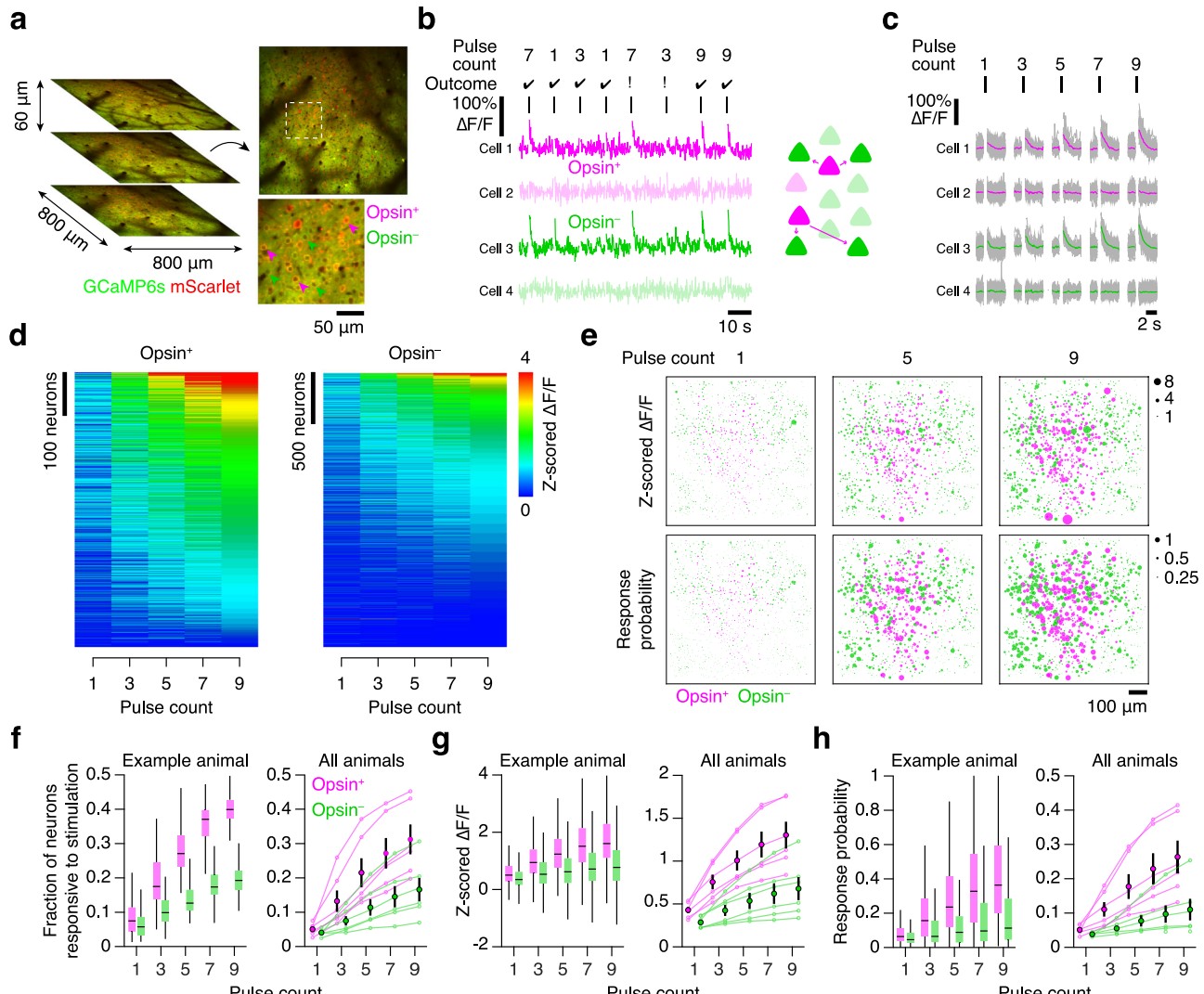

**Fig. 2 | Photostimulation response increases with pulse count. a** Volumetric two-photon imaging. Left, three planes (800-by-800 μm, 60 μm inter-plane distance) imaged simultaneously at 7 Hz; right, example plane with example opsin-expressing (magenta) and opsin non-expressing (green) neurons. **b** Photostimulation-evoked ΔF/F traces for 4 neurons across 10 consecutive trials. Vertical bars indicate stimulation epochs. Trial outcome indicates whether the animal made a correct (check) or incorrect (cross) behavioral response. Inset, proposed flow of activity from opsin-expressing to opsin non-expressing neurons. **c** ΔF/F traces for the 4 neurons in **b** showing mean response to each pulse count over a single session. Grey, individual trials; color, mean. **d** Response amplitudes (z-scored ΔF/F) of all opsin-expressing and opsin non-expressing neurons from one animal sorted by responses to the 9 pulse stimulus. **e** Spatial map of neural responses from one imaging plane. Size of dot indicates either response amplitude (top) or response probability (bottom) to 1, 5, or 9 stimulus pulses. **f** Fraction of neurons responsive to stimulation as a function of pulse count. Left, example animal ($n = 624$ opsin-expressing neurons, $n = 2640$ opsin non-expressing neurons; box plot denotes quartiles, whiskers denote 1st and 99th percentiles, horizontal line denotes median); right, all animals (small dot, median across all full intensity discrimination task sessions; large dot, grand mean; error bars, s.e.m.). **g** Same as **f** for response amplitude as a function of pulse count. **h** Same as **f** for response probability as a function of pulse count.

Supplementary Fig. 8b). Relative redness was also stable across days (Supplementary Fig. 8c, d).

To determine whether opsin expression was impacting cellular health or response properties, we tracked whisker movements ('whisking') in mice that learned the task ($n = 6$) and assessed whether whisking responsive neurons in the imaged population were impacted by opsin expression (Supplementary Fig. 9a, b). The fraction of whisking responsive neurons and their encoding scores were comparable between the opsin-expressing and opsin non-expressing populations (Supplementary Fig. 9c–e). Moreover, the fraction of whisking cells and their encoding scores remained stable over the course of training for both populations. We also examined activity during the inter-trial epoch ('spontaneous' activity; Methods), and found no difference between opsin-expressing and opsin non-expressing neurons over the course of training (Supplementary

Fig. 9f, g). Thus, observed changes in neural responsiveness are unlikely to be due to changes in opsin expression or physiological changes restricted to opsin-expressing neurons.

Stronger optical microstimulation is more detectable[26,27], so we next asked whether the amplitude of the evoked response early in training was greater among animals that learned the task. Specifically, we looked for differences in evoked responses between mice that learned ($n = 6$ mice, Supplementary Table 1) and those that did not learn ($n = 5$ mice) either early in training (days 1-3) or at the end of the time common to both learners and non-learners (late in training, days 6-8). We found that opsin-expressing neurons showed comparable response amplitudes between learners and non-learners early (learners: $0.63 \pm 0.24$, non-learners: $0.87 \pm 0.27$; $P = 0.151$, two-sided $t$-test comparing learners and non-learners) and late (learners: $0.35 \pm 0.18$, non-learners: $0.35 \pm 0.25$; $P = 0.991$) in training (Fig. 4g). There was also

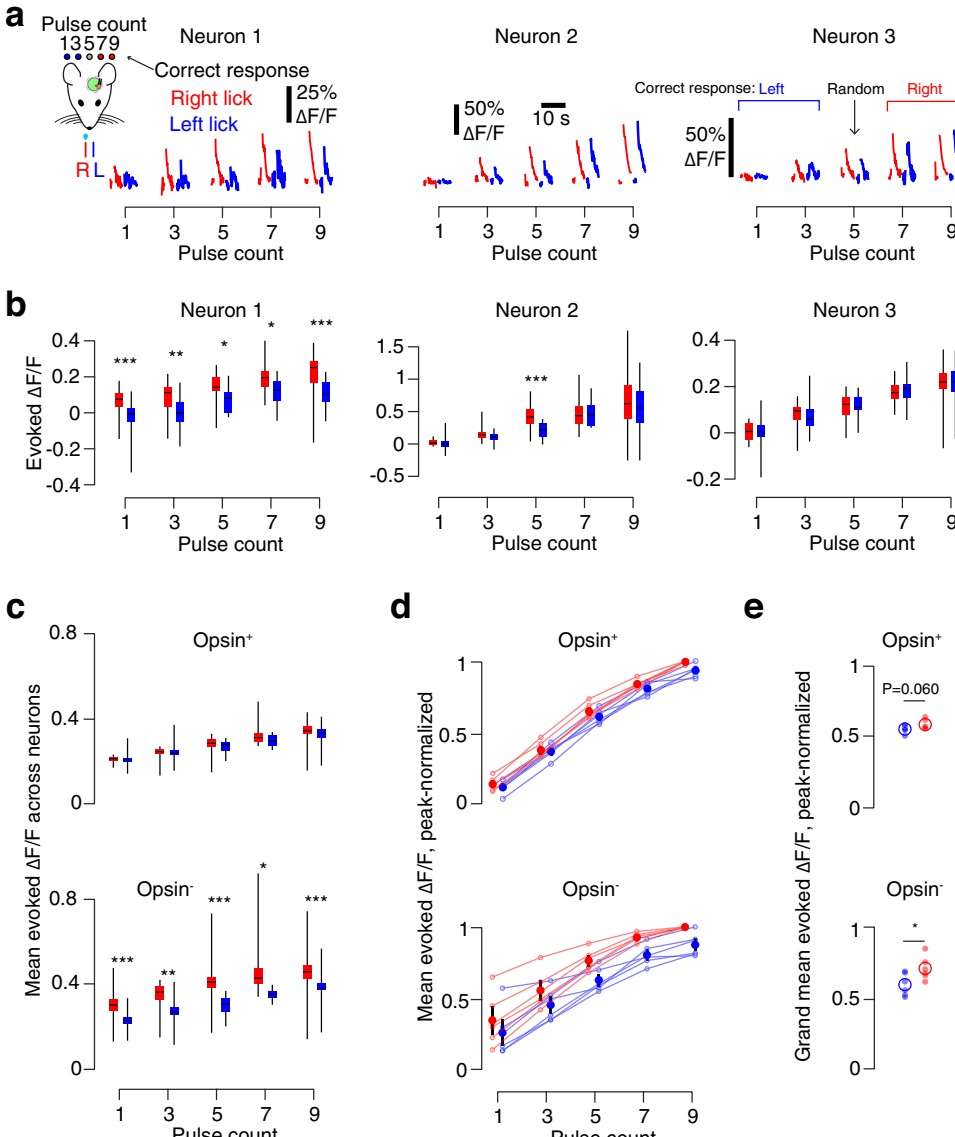

**Fig. 3 | Neural activity in vS1 predicts behavioral choice. a** Example neuron ΔF/F traces for all pulse counts, grouped by response. Red, animal licked right; blue, animal licked left. Left, neuron with choice-predictive activity for all pulse counts. Middle, choice-predictive activity on 5 pulse trials only. Right, neuron without choice-predictive activity. Activity is from an expert mouse in the final stage of the task. **b** Evoked ΔF/F across all trials in an example session for neurons in **a**. Box plot denotes quartiles, whiskers denote 1st and 99th percentiles, horizontal line denotes median. *$P < 0.05$, **$P < 0.01$, ***$P < 0.001$, two-sided $t$-test; unlabeled, $P \geq 0.05$. **c** Mean photostimulation-evoked ΔF/F across all neurons during an example session, restricted to timepoints prior to first lick. Box plots as in **b**. Top, opsin-expressing neurons; bottom, opsin non-expressing. **d** Peak-normalized photostimulation-evoked ΔF/F across all mice ($n = 6$, mice in final task stage). Thin line, individual mouse. Dot, mean across mice. Vertical line, s.e.m. **e** Peak-normalized photostimulation-evoked ΔF/F across all mice, averaged across all pulse counts. Small circle, single animal mean. Large circle, cross-animal mean. $P$-values labeled as in **a**, for two-sided paired $t$-test' $n = 6$ mice. Top, $P = 0.060$; bottom, $P = 0.017$.

no difference in response probability either early (learners: $0.17 \pm 0.04$, non-learners: $0.20 \pm 0.05$; $P = 0.242$) or late (learners: $0.11 \pm 0.05$, non-learners: $0.11 \pm 0.06$; $P = 0.798$) in training. Similarly, opsin non-expressing neurons showed no difference in response amplitude between learners and non-learners early (learners: $0.15 \pm 0.11$; non-learners: $0.08 \pm 0.03$; $P = 0.227$) or late (learners: $0.09 \pm 0.07$; non-learners: $0.04 \pm 0.01$; $P = 0.144$) in training (Fig. 4h). Response probability also did not differ early (learners: $0.05 \pm 0.04$; non-learners: $0.03 \pm 0.01$; $P = 0.320$) or late ($0.04 \pm 0.03$; non-learners: $0.03 \pm 0.00$; $P = 0.218$) in training. Thus, activity declined in both learners and non-learners and evoked activity was comparable across both cohorts.

### Learners show greater photoresponsive population stability
Even though aggregate responsiveness was similar between learners and non-learners across training, individual neurons may show

variation in their evoked activity that distinguishes learners from non-learners. Response variability within cortical neurons been observed in many cortical areas[13] and is thought to influence perception and learning[15]. Within a single training session, we observed neurons whose responses were stable across the session and others whose responses increased or decreased (Fig. 5a). At the population level, we computed the evoked ΔF/F (Methods) for individual 9-pulse trials (Fig. 5b) and obtained a vector of values (one per neuron) for a given trial. To quantify stability, we computed the correlation between evoked ΔF/F vectors across trial pairs (Fig. 5c). Periods of elevated trial-to-trial correlation were interspersed with periods of lower correlation, though the population response could be stable over many trials (Fig. 5d).

We next asked if representational stability differed among learners and non-learners. To obtain an overall measure of stability for a

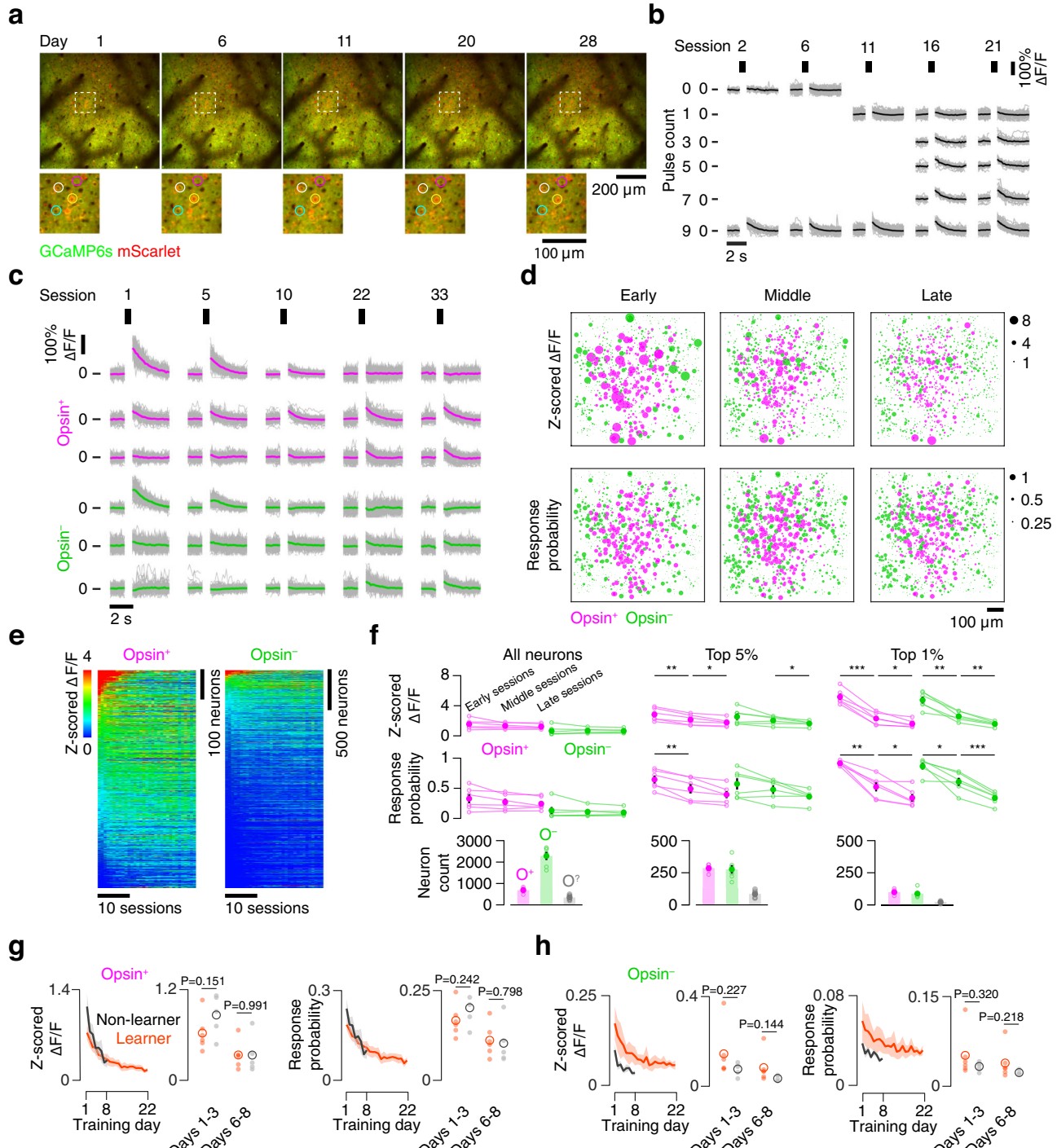

**Fig. 4 | Population dynamics during training. a** Example plane across multiple days. **b** Responses of an example neuron across training for each pulse count. Vertical bars, stimulation epoch. Grey, individual trials; black, mean. **c** Responses of example neurons on 9 pulse trials across training. Top three rows: opsin-expressing neurons (magenta) with decreasing (top), stable (middle), and increasing (bottom) responses across sessions. Bottom three rows: same for three opsin non-expressing neurons (green). Grey, individual trials; color, mean. **d** Spatial maps of neurons from one imaging plane at three time points along the training progression. Size of dot indicates either response amplitude (top) or response probability (bottom). **e** Response amplitude across sessions for all neurons from one animal. Neurons are sorted by the mean response amplitude over the first three training sessions. **f** Responsiveness across training. Left, all neurons (magenta, opsin-expressing; green, opsin non-expressing). Each point indicates the median response amplitude (top; z-scored ΔF/F) or probability of response (middle) of neurons from a single

animal across three early, middle, or late sessions. Large dot, grand mean across animals. Error bars, s.e.m. Bottom, the total number of opsin-expressing (O⁺), opsin non-expressing (O⁻), and expression-ambiguous (O?) neurons (Methods). Center, same as left panel but for neurons in the top 5% of response amplitudes (Methods). All neurons meeting the response criterion on at least one session are included and the same neurons are compared in each time bin. Right, same as left panel but for neurons in the top percentile of response amplitudes. *$P < 0.05$, **$P < 0.01$, ***$P < 0.001$, two-sided paired *t*-test; unlabeled, $p \geq 0.05$. **g** Change in response amplitude (left, z-scored ΔF/F) and response probability (right) for learners (orange) and non-learners (black) among all opsin-expressing neurons. Dark line, mean; envelope, s.e.m. Dots reflect the mean across individual mice for early days common to all animals (days 1-3) or late days (6-8), with hollow circle indicating the grand mean. *P*-value is given for two-sided *t*-test comparing learners and non-learners. **h** As in **g**, but for opsin non-expressing neurons.

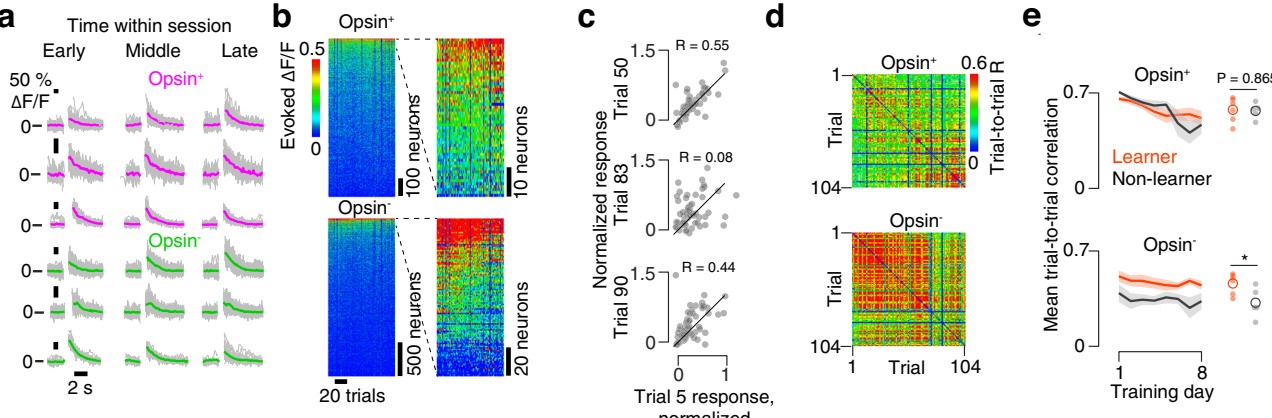

**Fig. 5 | Representational stability within a session. a** Example response to 9-pulse stimulus for several neurons at various points during the session. Magenta, opsin-expressing; green, opsin non-expressing. Top neuron, increasing response; middle, stable response; bottom, declining response. **b** Photostimulus-evoked ΔF/F for opsin-expressing (top) and opsin non-expressing (bottom) neurons for an example animal across all trials from a single session. **c** Pearson correlation between evoked ΔF/F response on an early trial and various subsequent trials. Each dot represents a neuron that was responsive at some point during the session. **d** Correlation between evoked ΔF/F vectors for all pairs of trials on an example imaging day. Top, matrix for opsin-expressing neurons; bottom, matrix for opsin non-expressing neurons. **e** Mean within-day trial-to-trial correlations during early sessions (1-8) in opsin-expressing (top) and opsin non-expressing (bottom) neurons. Black, non-learners; orange, learners. Thin lines, individual animals; thick line, mean across animals ($n = 6$ mice, learners; $n = 5$ non-learners). Small circle, individual animal mean; large circle, cross-animal mean. *$P < 0.05$, two-sided $t$-test comparing learners and non-learners. Top, $P = 0.865$; bottom, $P = 0.032$.

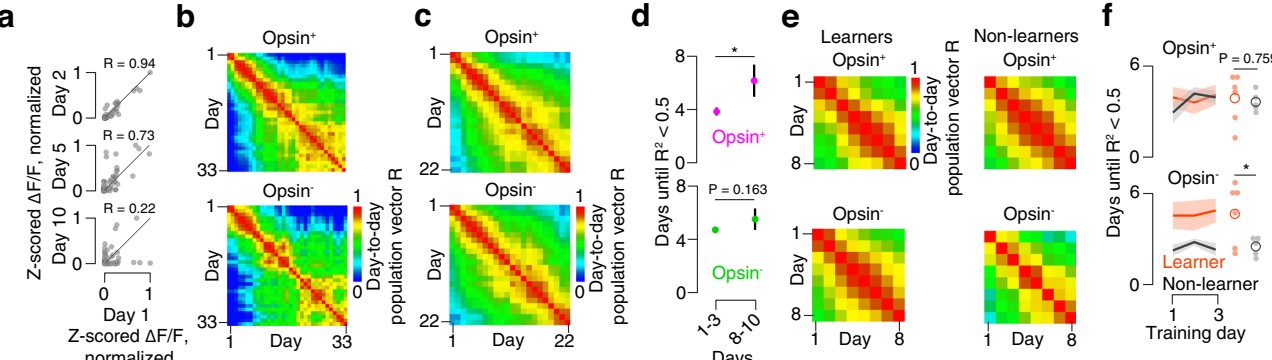

**Fig. 6 | Representational stability across sessions. a** Pearson correlation between mean evoked ΔF/F response on day 1 and example subsequent days for 9-pulse trials. Each dot represents a neuron that was responsive in at least one session. **b** Cross-day correlation matrix for an example mouse. Top, opsin-expressing neurons; bottom, opsin non-expressing neurons. **c** Average cross-day correlation matrix across all mice that learned the task for the first 22 days of imaging. **d** Number of days until $R^2$ drops below 0.5. Magenta, opsin-expressing; green, opsin non-expressing neurons. Dot indicates cross-animal mean, line indicates s.e.m. *$P < 0.05$, paired two-sided $t$-test comparing early to late day ranges, $n = 6$ mice.

Top, $P = 0.035$; bottom, $P = 0.163$. **e** Average cross-day correlation matrix across all mice for the first 8 days of imaging. Left, mice that learned; right, mice that did not learn. **f** Number of days from given starting point until cross-day $R^2$ drops below 0.5. Top, opsin-expressing neurons; bottom, opsin non-expressing neurons. Orange, learners ($n = 6$ mice); black, non-learners ($n = 5$ mice). Dark line, mean across mice. Envelope, s.e.m. Small circle, individual animal mean for days 1-3; large circle, cross-animal mean. *$P < 0.05$, two-sided $t$-test comparing learners and non-learners. Top, $P = 0.759$; bottom, $P = 0.048$.

given session, we computed the mean trial-to-trial correlation of response vectors for a given session in all mice, restricting our analysis to 9-pulse trials. Among opsin-expressing neurons, there was no difference in the mean trial-to-trial correlation across the training days common to all mice (days 1-8) between learners and non-learners (Fig. 5e; learners: $0.58 \pm 0.05$, $n = 6$ mice; non-learners: $0.58 \pm 0.11$, $n = 5$; $P = 0.865$, two-sided $t$-test). Among opsin non-expressing neurons, however, learners had higher mean trial-to-trial correlation than non-learners (learners: $0.47 \pm 0.02$, $n = 6$ mice; non-learners: $0.34 \pm 0.03$, $n = 5$; $P = 0.032$). Thus, among opsin non-expressing neurons, within-session photostimulus response variability is higher among non-learners.

We next examined representational stability across sessions. For each session, we generated a vector with the mean evoked ΔF/F across 9-pulse trials for each neuron. We then compared days by

computing the correlation between one day's vector and analogous vectors obtained for subsequent days. For sessions farther apart in time, correlations between these vectors declined (Fig. 6a, b). Aggregated across animals, the opsin-expressing population showed a decline in the rate of turnover over the course of training, whereas opsin non-expressing neurons exhibited steady drift (Fig. 6c). To quantify this, we measured the number of days it took for $R^2$ between two population vectors to fall below 0.5. In learners, this increased from $3.9 \pm 0.3$ days in early training (days 1-3) to $6.2 \pm 1.2$ days in middle training (days 8-10) for opsin-expressing neurons (Fig. 6d; $P = 0.035$, two-sided $t$-test, $n = 6$ mice), but remained unchanged among opsin non-expressing neurons (early: $4.6 \pm 0.2$ days; late: $5.4 \pm 0.8$ days; $P = 0.163$, two-sided $t$-test). Thus, inter-day response stability increased for opsin-expressing neurons as training progressed.

To determine whether inter-day response stability differed between learners and non-learners, we compared the two using the days for which we had data across all mice (days 1-8). Correlations between single day population response vectors were high for both learners and non-learners among opsin-expressing neurons but were lower for non-learners among opsin non-expressing neurons (Fig. 6e). We measured the mean number of days until $R^2$ between two population vectors fell below 0.5. For days with only a few subsequent training sessions common to both learners and non-learners (i.e., day 6), this condition was often never met. Therefore, we restricted our analysis to the days for which all animals yielded a value for this metric (days 1-3). We found no difference among learners and non-learners for opsin-expressing neurons (Fig. 6f; days until $R^2 < 0.5$, learners: $3.9 \pm 0.3$, $n = 6$ mice; non-learners: $3.8 \pm 0.7$ days, $n = 5$; $p = 0.759$, two-sided $t$-test). For opsin non-expressing neurons, however, $R^2$ remained above 0.5 longer for learners ($4.6 \pm 0.1$ days, $n = 6$ mice) than for non-learners ($2.4 \pm 0.3$ days, $n = 5$; $p = 0.048$, two-sided $t$-test). Thus, mice that fail to learn the task exhibit higher levels of both intra- and inter-day response stability among opsin non-expressing neurons.

### Stimulus decoding degrades more rapidly in mice that fail to learn the task

Representational stability is thought to impact downstream decoding of sensory cortical activity[15]. This suggests that the ability to decode the stimulus from neural activity may be different among learners and non-learners. We therefore compared decoding between learners and non-learners over the course of training. We evaluated decodability using receiver operating characteristic (ROC) analysis, discriminating between evoked ΔF/F on high (7, 9) and low (0, 1, and 3) pulse count trials. We quantified decoding efficacy by computing the area under the ROC curve (AUC; Methods) for each neuron on a given session. Stimulus decoding declined over the course of training in individual mice (Fig. 7a). Though transitions to more difficult task variants reduced decodability, decoding declined even between these transitions. In learners, the AUC among the top 5% of neurons by decoding ability on a given day (Methods) declined from $0.84 \pm 0.02$ (days 1-3) to $0.70 \pm 0.01$ (days 20-22) among opsin-expressing neurons (Fig. 7b; $P = 0.004$, $n = 6$ mice, two-sided $t$-test; comparing days 1-3 vs. days

20-22). Simultaneously, the fraction of opsin-expressing neurons with an AUC > 0.75 declined from $0.12 \pm 0.02$ to $0.02 \pm 0.01$ (Fig. 7c; $P = 0.003$). Smaller declines were observed among opsin non-expressing neurons, with the AUC among the top 5% of neurons dropping from $0.69 \pm 0.02$ to $0.61 \pm 0.01$ ($P = 0.044$, days 1-3 vs. days 20-22) and the fraction of neurons with AUC > 0.75 declining from $0.02 \pm 0.02$ to $0.01 \pm 0.00$ ($P = 0.006$). Thus, in mice that learned the task, stimulus decoding slowly declines over the course of training, despite steady or even improving performance.

We next asked whether decoder performance differed among learners and non-learners. We examined decoding on the first and last three common days (days 1-3 and 6-8, respectively). Initially (days 1-3), decoding among the top 5% of opsin-expressing neurons was higher for non-learners than learners (Fig. 7d; learners: $0.84 \pm 0.02$, $n = 6$; non-learners: $0.91 \pm 0.01$, $n = 5$; $P = 0.006$, two-sided $t$-test comparing learners with non-learners), with no difference among opsin non-expressing neurons (learners: $0.69 \pm 0.02$; non-learners: $0.65 \pm 0.02$, $P = 0.422$). The fraction of neurons with AUC > 0.75 was also comparable among learners and non-learners for both opsin-expressing (Fig. 7e; learners: $0.12 \pm 0.02$, $n = 6$; non-learners: $0.15 \pm 0.04$, $n = 5$; $P = 0.414$) and opsin non-expressing neurons (learners: $0.02 \pm 0.02$, $n = 6$; non-learners: $0.01 \pm 0.00$, $n = 5$; $P = 0.148$). For the last three common days (days 6-8), however, decoding for the top 5% of neurons was no longer significantly different among opsin-expressing neurons (learners: $0.82 \pm 0.02$; non-learners: $0.75 \pm 0.07$, $P = 0.168$), but became so among opsin non-expressing neurons, with non-learners exhibiting worse decoding performance (learners: $0.66 \pm 0.01$; non-learners: $0.59 \pm 0.02$, $P = 0.028$). Similarly, the fraction of neurons with AUC > 0.75 was comparable among opsin-expressing neurons in learners vs. non-learners (learners: $0.07 \pm 0.0$, $n = 6$; non-learners: $0.04 \pm 0.05$, $n = 5$; $P = 0.266$), but was lower among opsin non-expressing neurons (learners: $0.02 \pm 0.00$, $n = 6$; non-learners: $0.00 \pm 0.00$, $n = 5$; $P = 0.027$). Thus, stimulus decoding performance declines more rapidly in non-learners than learners.

## Discussion

We tracked neural activity in vS1 as mice learned to perform an optical microstimulation pulse count discrimination task (Fig. 1). Evoked

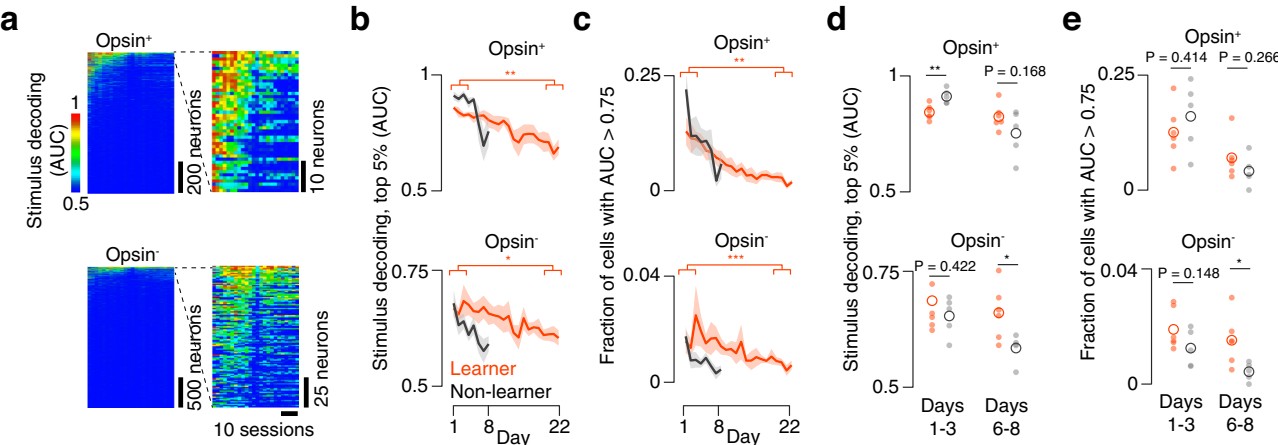

**Fig. 7 | Stimulus decoding across sessions. a** Decoding of stimulus across all neurons and sessions for an example mouse. Stimulus decoding for individual neurons was calculated as the area under the curve (AUC) for a linear decoder that must indicate if the stimulus contained a high (>5) or low (<5) pulse count (Methods). Top, opsin-expressing neurons. Bottom, opsin non-expressing neurons. Left, entire population, sorted by mean AUC across days. Right, top 5% by AUC.
**b** Stimulus decoding (AUC) among best-decoding 5% of cells up to the last common session for learners (orange, $n = 6$) and non-learners (black, $n = 5$). Dark line, mean; envelope, s.e.m. Top, opsin-expressing population; bottom, opsin non-expressing.

*$P < 0.05$, **$P < 0.01$, ***$P < 0.001$, paired two-sided $t$-test comparing mean across days 1-3 with mean of days 6-8. **c.** As in **b**, but for the number of neurons with AUC > 0.75. **d** Stimulus decoding (AUC) among top 5% of decoding cells up to the last common session for learners (orange, $n = 6$) and non-learners (black, $n = 5$) averaged across the first three common days (1-3; left) or last three common days (6-8; right). Small circle, single animal mean; large circle, cross-animal mean. $P$-value is given for two-sided $t$-test comparing learners and non-learners. **e** As in **d**, but for the number of neurons with AUC > 0.75.

activity in this task was proportional to the number of photostimulation pulses delivered (Fig. 2). Individual mice learned at different rates, with some mice failing to learn the task in the time provided. Among well-trained mice in the final task stage, evoked activity was predictive of animal choice, though only among opsin non-expressing neurons indirectly driven by photostimulation (Fig. 3). Over the course of training, evoked activity among the photoresponsive population declined (Fig. 4). Throughout training, the photoresponsive population exhibited a steady rate of neural turnover. Animals that exhibited more stable neural responses among opsin non-expressing neurons, both within and across sessions, were more likely to learn than those with less stable responses (Figs. 5, 6). Lower response stability coincided with greater declines in stimulus decoding, and mice that failed to learn exhibited poorer stimulus decoding in later training (Fig. 7).

Lower response stability and degradation in stimulus decoding were only predictive of learning among opsin non-expressing neurons. Thus, the activity directly evoked in cortex by photostimulation did not account for the rate of learning. However, the efficacy with which that activity drove consistent responses in downstream neural populations did impact perception[13–15]. Photostimulation produced more consistent responses among the directly stimulated population, as demonstrated by the higher correlations among the population response for opsin-expressing neurons both within and across sessions relative to opsin non-expressing neurons (Figs. 5e, 6f). Factors that could contribute to varied recruitment of downstream neurons include top-down modulation[42,43] as well as variable excitatory recurrence in downstream populations[33,37,44–46]. Our work suggests that the efficacy with which the stimulated population can propagate its activity to downstream neurons predicts learning ability.

Representational instability has been observed in many sensory[1–8] and motor[10,11,47,48] cortices, as well as hippocampus[12,49–51]. Under some conditions, however, sensory cortical[52–54], motor cortical[55–57], and hippocampal[58] representations can be stable for extended periods at the single neuron level. Factors that contribute to greater response stability include extended periods of task training[6,41,59], task engagement[49,50,60], and artificial versus naturalistic stimuli[3]. We observe representational instability in the context of direct cortical stimulation, even in highly trained animals engaged in the task. This supports the view that such instability is an intrinsic feature of cortex, though it is does not rule out the possibility that other structures contribute to the observed instability[42].

We observe a large degree of sparsification in the photoresponsive population over the course of the first week of training, both in terms of the size of the responsive population and the typical response for a given neuron. With novel natural stimuli, both increases and decreases in responsive populations have been observed. For example, novel, behaviorally-relevant tones become overrepresented across learning in auditory cortex[61,62], whereas whisker representations can shrink following environmental enrichment in vS1[63]. Direct stimulation of cortex or of cultured cortical neurons can drive both increases[23,64–66] and reductions[67,68] in responsiveness. Candidate mechanisms for declining responsiveness include increased inhibitory input onto pyramidal neurons[69], reduced excitability[70,71], and reductions in excitatory connectivity[67]. Reduced excitability is a less likely mechanism in our experiment, given the lack of change in whisking responses and stability of spontaneous activity (Supplementary Fig. 9), which should both be impacted if global excitability changed. Given that the most active neurons experience disproportionately larger response declines, sparsification is unlikely to be due to global mechanisms, such as a general increase in inhibition. Instead, cell-specific changes likely account for the observed dynamics, such as targeted changes in inhibition.

Microstimulation has been used extensively in the study of perception[17,18], yet tracking the dynamics of microstimulation-evoked activity over learning has proven difficult. Chronic electrical microstimulation poses numerous challenges: it is invasive, can inadvertently activate neurons up to millimeters away from the stimulation site[28], and can cause tissue degradation at the site of electrode insertion[29]. Furthermore, simultaneous recording is complicated by a prominent stimulation artifact[30], and even studies with concurrent recording rarely examine the activity evoked at the stimulation site[72–75] (but see ref. 76) Finally, electrical microstimulation cannot target genetically separable neural populations. Optical microstimulation overcomes these issues. Photostimulation using an external LED does not require an intracortical implant, the stimulation is spatially and genetically confined, and calcium imaging allows us to observe evoked responses in thousands of neurons at single cell resolution[33,34]. In contrast to objective-based optogenetic illumination, our LED-based approach also allows the objective to move while maintaining consistent stimulation at the target site. This allows for imaging at multiple depths and other brain areas and permits training away from the microscope.

We show that over the course of optical microstimulation task training, evoked activity in the stimulated region of vS1 exhibits instability both within and across sessions. We find that greater stability in optogenetically evoked activity is observed among mice that successfully learn to use evoked activity to perform a task. Thus, learning is associated with more stable sensory cortical representations.

## Methods
### Animals and surgery
Adult Ai162 (JAX 031562) X Slc17a7-Cre (JAX 023527)[36] mice (15 male, 1 female; Table S1) were used throughout. These mice express GCaMP6s exclusively in cortical excitatory neurons in a tetracycline transactivator-dependent manner. To suppress expression during development, breeders were fed a diet that included doxycycline (625 mg/kg doxycycline; Teklad). Mice received doxycycline until weaning. All animal procedures were in compliance with protocols approved by New York University's University Animal Welfare Committee.

Mice (6-10 weeks old) were anesthetized with isoflurane during viral injections, surgical implantation, and LED placement (3% induction, 1.5% maintenance). A titanium headbar was attached to the skull with cyanoacrylate (Vetbond). A circular craniotomy (3.5 mm diameter) was made in the left hemisphere over vS1 (center: 3.3 mm lateral, 1.7 mm posterior from bregma) using a dental drill (Midwest Tradition, FG 1/4 drill bit).

Following the craniotomy, virus encoding the soma-localized opsin ChRmine and the red fluorophore mScarlet (AAV-8-CaMKIIa-ChRmine-mScarlet-Kv2.1-WPRE, 2.48 × 10¹³ vg/mL, diluted 1:500 in 1X PBS; generously provided by Dr. Karl Deisseroth) was injected into vS1. A glass capillary (Wiretrol II, Drummond) was pulled using a micropipette puller (P-97, Sutter Instrument) and the tip beveled to 25° with a tip diameter of 25 µm. The pipette was back-filled with mineral oil (M5904, Sigma-Aldrich) and 2 µL of viral solution was drawn into the tip. Three 100 nL injections were made 300 µm below the dura and spaced 400 µm apart in a triangle centered on the typical anatomical location of the C2 barrel. For each injection, the pipette was lowered into the brain at a rate of 300 µm/min, followed by a 1-minute pause after which virus was injected at a rate of 20 nL/min using a hydraulic micromanipulator (Narishige MO-10). Following a 2 minute pause, the pipette was withdrawn at a rate of 300 µm/min with an additional 1 minute pause at a depth of 150 µm below the dura. Finally, the dura was removed using a pair of fine forceps (Fine Science Tools) and a double-layer cranial window (4.5 mm external diameter, 3.5 mm inner diameter; #1.5 coverslip; adhered with Norland 61 UV glue) was placed over the craniotomy. The cranial window and headbar were affixed to the skull with dental acrylic (Orthojet, Lang Dental).

Following surgical recovery, mice were trimmed to whiskers C1-3 and placed on water restriction, with regular trimming thereafter. To confirm that the area of opsin-expression fell within vS1, the location of the barrels corresponding to the C1-3 whiskers was identified by measuring touch-evoked GCaMP6s ΔF/F at coarse resolution during an imaging session during which the mouse was awake but not performing any task (4X objective; 3 × 3 mm field of view; Supplementary Fig. 1a–c).

LED placement was performed 2-4 weeks after injection. LEDs (590 nm, LXZ1-PL01, Lumileds) were connected to the LED driver by soldering 3 cm of polyurethane enameled copper wire (34 AWG) to each pad of the LED. A 2-pin, flat flex cable connector (Digikey) was then soldered to the free ends of the copper wire and secured with epoxy (Devcon). To waterproof the LED, a thin layer of clear nail polish (Sally Hansen) was applied to all surfaces of the LED. LEDs were then affixed to the animals' cranial windows. Following anesthesia, a small arc (60°) of dental cement was drilled away on the anterior-medial edge of the craniotomy to expose the edge of the cranial window. Cyanoacrylate was applied to the drilled area to ensure that the craniotomy remained sealed. The LED was placed at a 30° angle on the edge of the cranial window with the most lateral edge of the LED 0.5-1 mm away from the center of the opsin-expressing area (Supplementary Fig. 1). The medial edge of the LED and the copper wires were secured to the dental cement and the LED connector secured to the posterior edge of the headbar using cyanoacrylate. Waterproofing was confirmed by placing water on the cranial window and ensuring that no current passed between the water and the LED.

### Light propagation model

The LED was modeled as an 8 × 8 grid of point sources positioned in water above the cranial window (Supplementary Fig. 1e). A light ray from each point source thus passed through the following: water, interface between water and window glass, glass, interface between glass and brain, and brain tissue. At each interface, Fresnel equations were used to calculate the fractional transmission of light. Within the brain, scattering of light was accounted for using empirical measurements[77]. The power density at different depths within the brain was computed by scaling the max power at the LED face (measured using a photodiode) by the fractional intensity, fractional transmission at each interface, and fraction of light remaining after scattering. This quantity was divided by the surface area of a half-sphere to yield the predicted power density at a given location in the brain. Light radiation from each point source was treated independently and averaged to compute the final predicted power density.

### Photostimulation system

Optogenetic stimulus delivery was controlled by a LabJack T7, with commands originating from a MATLAB user interface on a separate computer. The stimulation and masking flash LEDs were controlled using LED drivers (T-cube, ThorLabs) whose output currents were controlled by the LabJack. The signal to the stimulation LED was terminated using a 4-pin flat flex cable connector (Digikey) that was mated to the LED connector on the animal during behavior. The masking flash consisted of 3 LEDs (595 nm, XPEBAM-L1-0000-00A01, Cree LED) that were spectrally matched to the stimulation LED and placed near the animal's face to illuminate the eyes.

### Behavior

Mice were trained on an optogenetic pulse count discrimination task in which the number of light pulses from the stimulation LED was associated with a water reward from one of two lickports (Fig. 1b). Each trial consisted of three epochs: stimulus (500 ms), delay (500 ms), and response (<2 s) (Fig. 1c). During the stimulus epoch, between 0 and 9 light pulses (5 ms, 20 Hz) were presented from the stimulation LED. Pulses were randomly distributed among 9 time bins (5 ms, 20 Hz) to

prevent the animal from using timing cues to solve the task (Fig. 1d, Supplementary Fig. 4). Masking flash pulses (15 ms, 20 Hz, starting 5 ms prior to the optogenetic stimulus pulse and ending 5 ms after) were presented at each time bin on all trials to prevent the animal from using visual cues to solve the task. The PMT shutters were closed 50 ms before the start of the first photostimulus time bin and opened 50 ms after the end of the last bin (total closure time: 505 ms). Thus, the masking flash provided a visual cue for the onset of the stimulus epoch, whereas the shutter provided an auditory cue. Following the stimulus epoch, there was a 500 ms delay after which an auditory cue (3 kHz, 50 ms) signaled the beginning of the response epoch. At this point, mice could indicate their decision by licking one of two lickports spaced 3 mm apart along the medial-lateral axis. The lickports were moved into reach during the response epoch by electrical motors. Well-trained mice withheld licking until the lickport came into reach. Correct responses were rewarded with 3-7 μL of water which the animal could collect for 1-2 s. Incorrect responses were punished with immediate withdrawal of the lickport and a timeout. After the response epoch, there was a 7 s inter-trial interval.

Training proceeded in a standard sequence. Following LED placement, water-restricted mice were handled and head-fixed to habituate animals to the behavioral apparatus. Animals were first trained on a photostimulus detection task in which 9 stimulation pulses predicted reward from the right lickport and 0 pulses predicted reward from the left lickport (Fig. 1b, e). Mice were evaluated using a sliding 61-trial window over which both percent correct (Fig. 1e) and d-prime were calculated. For each session, a peak d-prime across these sliding windows was computed. Mice that had attained at least two sessions with a peak d-prime of 1.5 and continued to have peak d-prime greater than 1.5 by the 8th session were considered learners; mice that had not yet attained such performance, or had regressed despite two consecutive days of meeting criteria, were considered non-learners. As mice attained our performance criterion (peak d-prime over a 61 trial window exceeding 1.5) for more than 2 consecutive sessions for a given stage, the task progressed as follows: 0 pulses (left) vs. 9 pulses (right); 1 pulse (left) vs. 9 pulses (right); 1 and 3 pulses (left) vs. 7 and 9 pulses (right); 1, 3, and 5 pulses (left) vs. 5, 7, and 9 pulses (right). 5 pulse trials were randomly and equally rewarded between left and right. To maintain high performance on the final task and reduce bias, we modified the probability of presentation of each pulse count such that 1 and 9 pulse trials occurred up to 8x more frequently than intermediate pulse counts.

The behavioral task was controlled by a BPod state machine (Sanworks) and custom MATLAB software. The auditory response tone was controlled by a low-latency audio board (Bela). Lickport motion was controlled by a set of 3 motorized actuators (Zaber) and an Arduino. Licks were sensed using a custom electrical detection circuit[78] (Janelia).

### Whisker videography and whisking representation analysis

Whisker video was acquired using custom MATLAB (MathWorks) software from a CMOS camera (Ace-Python 500, Basler) running at 400 Hz and 640 × 352 pixels and using a telecentric lens (TitanTL, Edmund Optics). Illumination was provided by a pulsed 940 nm LED (SL162, Advanced Illumination). 7 s of each trial were recorded, starting 500 ms prior to shutter closure and including both the microstimulation and response periods. Data was processed on NYU's High Performance Computing (HPC) cluster. First, candidate whiskers were detected using the Janelia Whisker Tracker[79]. Next, whisker identity was refined and assessed across a single session using custom MATLAB software[6,45]. Following whisker assignment, the angle (θ) for one whisker (typically C2) was calculated, with protraction yielding more positive θ values, by convention.

Whisking classification was performed using a per-neuron generalized linear model (GLM) that predicted ΔF/F from the whisker

angle, $\theta$[6,45,80]. The whisking model fit excluded the ~2.5 s starting 1 s prior to shutter closure and ending 1 s following shutter closure, thereby excluding any photostimulation-related whisker movements. An encoding score was assigned to each neuron by computing the Pearson correlation between the model-predicted $\Delta F/F$ and actual $\Delta F/F$. Neurons for which this value exceeded 0.15 were assigned to the touch and/or whisking representations. Whisking scores were computed for each day of imaging.

## Two-photon microscopy

Calcium imaging was performed using a custom two-photon microscope (http://openwiki.janelia.org/wiki/display/shareddesigns/MIMMS). The microscope consisted of a 940 nm laser (Chameleon Ultra 2, Coherent), a Pockels cell (350-80-02, Conoptics), two galvanometric scanners (6SD11268, Cambridge Technology), a resonant scanner (6SC08KA040-02Y, Cambridge Technology), a 16x objective (N16XLWD-PF, Nikon), an emission filter for green fluorescence (FF01-510/84-30, Semrock), an emission filter for red fluorescence (FF01-650/60, Semrock), and two GaAsP PMTs (H10770PB-40, Hamamatsu). Each PMT had an associated shutter (VS14S1T1, Vincent Associates) that was controlled by a voltage signal from the LabJack.

Imaging data was acquired using Scanimage (Vidrio Technologies). Three 800-by-800 μm imaging planes axially spaced 60 μm apart were acquired at a rate of ~7 Hz. The objective was moved axially (total depth, 180 μm) with a piezo (P-725KHDS, Physik Instrumente). Power was depth-adjusted by the acquisition software with an exponential length constant of ~250 μm.

Imaging data were processed on the NYU High Performance Computing cluster using a semi-automated software pipeline that included image registration, segmentation, neuropil subtraction, $\Delta F/F$ computation, and calcium event detection[6]. After the first imaging day, motion-corrected mean images were collected for each plane and used as references during imaging on subsequent days. Alignment across days was performed as previously described[11].

Opsin expression was measured using mScarlet fluorescence. We defined opsin-expressing, opsin non-expressing, and ambiguous neurons using imaging data collected in the penultimate session for each animal. The green fluorescence signal was linearly demixed from the red fluorescence signal. Using the demixed pixel values, we calculated a 'redness score' as the mean pixel intensity across all pixels used for that neuron. We also computed the fraction of pixels with redness exceeding a noise threshold for each neuron following de-mixing. For each mouse, we manually selected a combined mean pixel value and pixel fraction threshold above which neurons were considered opsin-expressing. Neurons with a mean redness and fraction of red pixels below a second, lower set of thresholds were considered opsin non-expressing. Neurons with an intermediate redness score and fraction of red pixels were considered ambiguous and were excluded from analysis.

## Cortical lesions and histology

Cortical lesions were performed with either an 800 nm (Chameleon Ultra 2, Coherent) or 1040 nm (Fidelity HP, Coherent) laser by focusing the laser at 200-300 μm depth for 12-40 s at 100% power (1.6 W; Fig. 1h) at the infection site. Between 2 and 8 sites spaced 200-400 μm apart were targeted within the opsin-expressing area[81]. Lesion success was visually confirmed by an increase in GCaMP6s fluorescence in the targeted area. In 2 animals, a second round of lesions was conducted the subsequent day because opsin-expression and evoked activity was still evident. Animals were awake but not performing the task during lesioning and were monitored for signs of distress or discomfort. Imaging data was not collected following lesioning. Animals were perfused after completion of training. Brains were sectioned using a microtome (Leica), mounted on glass slides, and imaged on a fluorescent light microscope (VS120, Olympus). Histology revealed that typical lesions removed most, but not all, opsin-expressing neurons (Supplementary Fig. 6).

## Quantifying responsiveness

For analyses of responsiveness (Figs. 2–4), neurons were classified as responsive or non-responsive in every trial by comparing the post-stimulation $\Delta F/F$ to the baseline $\Delta F/F$. Baseline $\Delta F/F$ was selected to be the $\Delta F/F$ for the ~5.5 s (39 frames) preceding shutter closure on each trial. The evoked $\Delta F/F$ was calculated as the mean $\Delta F/F$ of the two frames immediately following shutter reopening. Neurons were considered photoresponsive on a given trial if the evoked $\Delta F/F$ exceeded the 97.5th percentile or fell below the 2.5th percentile of the distribution of baseline $\Delta F/F$ values across all trials in a session. To compare responses between neurons and across sessions, we z-scored the post-stimulation $\Delta F/F$, using the mean and standard deviation of the baseline $\Delta F/F$ over all trials for a neuron in a given session. We identified neurons highly responsive to photostimulation by computing the mean z-scored $\Delta F/F$ over 9-pulse trials for all neurons in a given session, yielding one value per neuron per session. Neurons were labelled highly responsive if they exceeded either the 95th or 99th percentiles of this distribution on at least one session of the sessions being considered. Spontaneous activity was measured using data excluding the period 1 s before to 5 s after any photostimulation. The calcium event rate was obtained using a template matching algorithm[6]. In cases where evoked response levels are reported, only the two frames immediately after shutter opening are used, so that reported evoked activity precedes any licking.

## Within- and cross-day representational stability analyses

To quantify within-day representational stability (Fig. 5), we first measured the evoked $\Delta F/F$ across all neurons for 9-pulse trials. For each 9-pulse trial, we constructed a population response vector consisting of the evoked $\Delta F/F$ for each neuron. The Pearson correlation between all pairs of population response vectors was then computed, yielding an inter-trial correlation matrix. Analysis was performed separately for opsin-expressing and opsin non-expressing populations. To compare mice that learned with those that did not, we first computed the mean of this matrix, using the upper triangular portion of the matrix excluding the diagonal. The comparison was made using the mean of this value across the training days that were common to all mice in the dataset (days 1-8).

To assess cross-day representational stability (Fig. 6), we first computed the mean evoked $\Delta F/F$ across all neurons on 9-pulse trials. This resulted in a single population response vector consisting of the mean evoked $\Delta F/F$ for each neuron for that session. The Pearson correlation between all pairs of population response vectors was then computed, yielding a cross-day correlation matrix. To compare mice that learned with those that did not, we used this matrix to calculate the number of days from each day until $R^2$ fell below 0.5 (i.e., the day beyond which a given day's population response vector could no longer account for at least half of the variance for subsequent days' population response vector). We only examined the days for which all mice had a valid value for this number (days 1-3).

## Stimulus decoding analysis

Stimulus decoding was assessed using receiver operating characteristic (ROC) analysis. For each day, we partitioned the trials into high (7, 9 pulse) and low (0, 1 and 3 pulse) stimulus intensity trials, excluding 5 pulse trials on days where they were present. For each trial, we obtained the evoked $\Delta F/F$. ROC analysis was performed by sliding a criterion threshold through the range of evoked $\Delta F/F$ values across the two trial types, thereby classifying responses as false alarms ($\Delta F/F >$ threshold on low pulse count trial) or hits ($\Delta F/F >$ threshold on high pulse count trial). We report the area under the curve (AUC) resulting from this analysis[82]. Two metrics were employed to assess decoding

efficacy: first, we computed the mean AUC across the top 5% of AUC values for a given day; second, we examined the number of neurons having an AUC exceeding 0.75 on any given day. In all cases, statistical comparisons were made by taking the average of one of the aforementioned metrics across three days. For comparing early to late decoding among mice that learned the task, we compared days 1-3 to the last three days common to all ($n$ = 6) mice that learned: days 20-22. For comparisons between learners and non-learners, we used days 1-3 and the last three common days, days 6-8.

## Statistics and reproducibility
For comparisons between samples, two-sided paired and unpaired $t$-tests were used. For correlation tests, Pearson's correlation was used to identify a linear correlation coefficient (R) and test for significance.

## Reporting summary
Further information on research design is available in the Nature Portfolio Reporting Summary linked to this article.

## Data availability
The data generated in this study are available at http://peronlab.org/data/2023_pancholi_microstim_learning.zip.

## Code availability
Source code used in this paper is available at http://github.com/peronlab/2023_pancholi_natcomms.

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

## Acknowledgements

We thank Bettina Voelcker, Keelin O'Neil, and Aaron Lanz for discussion and technical assistance. We thank Sadrah Sadeh, Claudia Clopath, Robert Froemke, Anthony Movshon, and Michael Long for discussion. This work was supported by the Whitehall Foundation and the National Institutes of Health (RO1NS117536 for R.P., L.R., and S.P.; F31NS120483 for R.P.; T32GM007308 for L.R. and R.P.).

## Author contributions

R.P. and S.P. designed the study. R.P. carried out the experiments. L.R. performed histology and developed the lesion method. R.P. and S.P. performed data analysis and wrote the paper.

## Competing interests

The authors declare no competing interests.
