## [Peer Review File · Nature Communications]

Learning in a sensory cortical microstimulation task is associated with elevated representational stabilityREVIEWER COMMENTS

Reviewer #1 (Remarks to the Author):

In this manuscript, Pancholi et al. examine whether the stability of cortical representations across sessions is related to learning of a task involving perception of artificial percepts. This is an important research question - although a number of labs have described the phenomenology of representational drift, the impact on learning and memory is not well understood. The authors took an interesting approach, forgoing the early sensory system entirely by directly stimulating random subsets of excitatory cortical neurons in vS1. The experiments appear to be carefully done, the analyses are clear and convincing, and the manuscript is very well written. My only major gripe, as I discuss below, is that the conclusions at times outstrip the data, and I would recommend some revision of the discussion. I also had a few minor concerns about the experimental details. Overall, however, these results will be a welcome addition to the field. Specific comments below:

1) The authors find that mice that learned the task show greater stability of cortical representations across sessions compared to non-learners, and use this result to conclude that “representational stability fundamentally constrains perception”. However, this result is only correlative, and there are other interpretations of the data. For example, when the learning animals understand the contingencies of the behavioral task at some higher level, feedback to sensory cortex might act to stabilize sensory representations. This could lead to the exact same result without representational stability explicitly constraining behavioral perception. Indeed, I suspect sensory perception constrains behavior much less than simply understanding how to associate the different pulse numbers to arbitrary motor actions, which likely occurs in ALM or other regions outside of vS1.

I still find the results interesting and think they will be of note to the field, but I do think the abstract/discussion should be revised to consider these possibilities and dial back the claims to those that are fully supported by the data.

2) I was intrigued by the authors’ approach of measuring representational stability by direct cortical stimulation. However, one worry I had about their approach was that exciting large numbers of nearby excitatory neurons could generate localized epileptiform activity which might interact with their experimental measures. Did the authors test in any way for the presence of hypersynchronous interictal activity?

3) In a number of cases, the authors restricted analyses to the top 5th percentile of responsive neurons. It would be helpful to carry out the same analyses for all responsive neurons in each of these cases.

Selecting one tail of the distribution can cause iceberg effects and either enhance or obscure observed effects.

4) The authors used almost exclusively male mice (except for one lone female). NIH policy is to use a sex-balanced sample unless there is a clear scientific justification. Maybe I missed it, but was there any justification for using such a biased sample?

Minor typos/corrections:

Line 515: extra "X" in the the JAX number for SLc17a7-Cre

Line 625: please cite prior art or describe the circuit

Line 639: extra "a" between "from" and "the"

Reviewer #2 (Remarks to the Author):

Pancholi and co-workers investigate chronic remodelling of sensory neural representations in mice that perform a learning paradigm. In this discrimination task, optogenetic stimuli are directly delivered to populations of cortical neurons, serving as cues. This elegant experimental design explores activity dynamics of neuronal populations with concurrent behavioural testing and allows reliable investigations of cortical function in a relevant behaviour. Mice show heterogeneous learning capabilities and are categorized as learners and non-learners. The authors compare representational remodelling in cortical neurons and report as main finding that animals which successfully learned the task show a higher representational stability than mice that do not learn the discrimination task in the time provided. They conclude that "the rate at which sensory cortical representations change constrains learning and impacts behavior".

The experiments are technically very demanding and of high standard, the analyses appear careful and thorough. The authors are congratulated for this effort. Also, the topic and question are timely and of high general interest. However, while reading the manuscript, I had problems in following the logic and was left unsure if their data really supported their conclusions. As such, I do not think this manuscript, at least in its current form, is suitable for publication in Nature Communications.

Main concerns:

1) The title and abstract claims that there is a causal relationship established between the stability of representations that underlies the ability of a mouse to learn or not. I do not think this study really allows to differentiate if the learning fails due to more unstable representations or the representations fail to stabilize due to the inability to learn. It would be safer to talk about a correlation between these measures and not to suggest any causal relationship.

2) The study design, using direct optogenetic activation of the cortex as cues, can offer advantages compared to using sensory stimuli (which are processed then in parallel in multiple brain areas) and in particular to electrical microstimulation. Electrical microstimulation is supposedly rather unstable due to gliosis, as outlined in the introduction.

However, I am not convinced that the optogenetic stimulation (at least as used in this study) is really much better. The design of this study (and other direct stimulation studies) makes the implicit assumption that the set of stimuli that is given to the brain always leads to the same set of activation patterns in the directly stimulated, “first order” neurons. The nice thing here is that the “input” to the brain is always the same and stable.

However, I am not sure that this is what the data shows. I have the impression that during the time course of the experiment a substantial adaptation of the “first order” neurons occurs and therefore leads to a constantly changing “input” to the brain, that the mice then are supposed learn and discriminate. Although not directly quantified, I think that Fig. 4e left (although this plot then should be re-sorted for each session), Fig. 4g (although here only top 5% are shown) and most strikingly Fig. 7a top (although here only the decoding power per cell is shown), indicate that the “first order“-activity pattern that is imposed by a given stimulus changes massively throughout the experiment. If I am not completely misinterpreting Fig. 7a, it says that at the end of the experiment, there is hardly any Opsin+ neuron left where the high pulse and the low pulse patterns induce any differential activity. Here, we are talking about the directly stimulated neurons!

This point must be clarified and directly addressed. A quantification as shown in 2F (which I guess is from the beginning of the experiment) should be shown for all the time points throughout the experiment. If my suspicion is right, I am afraid that the magenta curve would be essentially flat and low in the last sessions. Also, in Fig. 4g, this adaptation in the top 5% seems to be quite different throughout the initial training and therefore could be a major factor in determining what is a learner and a non-learner. Here, the authors decided to test for differences in the average throughout all sessions not revealing any difference between learners and non-learners. However, a test design as used in 7b-e comparing start and end would likely be significant. Also, it could be interesting to patch the cells in brain slices to see if the opsin-expressing, stimulated neurons would show strong changes in their intrinsic excitability.

Together, this was really the elephant in the room, when reading the manuscript. Also, the interpretation of their findings would probably change a lot, when also considering the stability of the input pattern as a crucial parameter that can impact on learning. It appears plausible that it is more difficult to learn to discriminate a moving target as a stable one.

3) The third point goes along a similar line: What could be important factors, that differentiate learners from non-learners (apart of the correlation with the stability of activity patterns in non-opsin neurons and eventually faster or slower rundown of imposed activity patterns in “first order” neurons)? Classically, one would consider the similarity or dissimilarity of activity patterns that are elicited by the different cues as a prime parameter how easy it is for the mouse to also discriminate behaviourally. Admittedly, often it is possible to decode the different stimuli from neuronal activity already before training, so that there would be no “need” to change the representations, but it has been shown to nevertheless occur (e.g. Huber, ..., Peron, et al, 2012, Nature; Poort et al., 2015, Neuron) and it would be therefore something obvious to look at. This aspect is touched upon in the very last figure quantifying the single-cell decoding ability. But in most of the manuscript’s figures only the activity patterns evoked by the 9 pulse stimuli are shown and quantified. I think it is important to provide also more detail on the structure of the activity patterns evoked by the other stimuli. Perhaps it is the stability of activity pattern evoked by the 1 pulse stimulus, that better differentiates learners from non-learners? Or it is the structure of the various activity patterns, and not the overall count of elicited activity. I would not be surprised, that some sort of normalization of overall activity is happening during the course of the experiment and the differentiation in the last sessions is based on the similarity or dissimilarity of the patterns.

Minor points:

The binary categorization of mice is somewhat arbitrary. Usually, most mice learn if you give them enough time. A non-binary, parametric correlation of learning (at least during the first 8 days) and the various neurometrics can be considered.

To better understand the learning process, it would be interesting to show not only overall performance, but also the hits, misses, FA, CR for the two categories of stimuli. I could imagine that there is some asymmetry in learning: The 9 pulse stimuli are learned initially and when introducing the 1 pulse stimulus the overall performance is mainly due to reduce the rate of FA.

I am surprised that many mice still perform the task above chance after the lesion. When removing all opsin expressing neurons, the mice should have no way to discriminate the trials. Was the lesion incomplete or what is the authors’ interpretation?

For the cited literature: Representational drift also has been recently shown in auditory cortex (Aschauer et al., 2022, Cell Reports), and earlier as in the cited studies in visual cortex (Montijn et al., 2015 Cell Reports; albeit not that explicitly named).

In Fig. 1E, I would put the scheme how the stimuli are introduced on top and somehow graphically fuse it better to the learning curve. For a long time, I associated the scheme with panel F. Also, one could use the same shapes in the scheme and in F.

In Fig. 3C there is an average trend towards more activity in right lick trials. Why is that? A matter of the hemisphere or of the contingency? I guess the interpretation is that in those trials where more activity as average is elicited, this may be easier confused with a right lick trial. But the actual discrimination of the stimuli may not depend on the overall rate of activity, despite the fact that higher activity rates are correlated with higher pulse numbers.

In Fig. 5D, it is not clear what is plotted at the bottom of the correlation matrices. The average of the columns? What would that tell us? How representative a given trial is compared to all others? Perhaps expand the legend a bit more. Also, what are the trials with very low correlations in the matrices?

Fig. 6, this analysis considers the response patterns to the 9 pulse stimulus? Please indicate clearly in legend.

REVIEWER COMMENTS

Reviewer #1 (Remarks to the Author):

In this manuscript, Pancholi et al. examine whether the stability of cortical representations across sessions is related to learning of a task involving perception of artificial percepts. This is an important research question - although a number of labs have described the phenomenology of representational drift, the impact on learning and memory is not well understood. The authors took an interesting approach, forgoing the early sensory system entirely by directly stimulating random subsets of excitatory cortical neurons in vS1. The experiments appear to be carefully done, the analyses are clear and convincing, and the manuscript is very well written. My only major gripe, as I discuss below, is that the conclusions at times outstrip the data, and I would recommend some revision of the discussion. I also had a few minor concerns about the experimental details. Overall, however, these results will be a welcome addition to the field. Specific comments below:

1) The authors find that mice that learned the task show greater stability of cortical representations across sessions compared to non-learners, and use this result to conclude that “representational stability fundamentally constrains perception”. However, this result is only correlative, and there are other interpretations of the data. For example, when the learning animals understand the contingencies of the behavioral task at some higher level, feedback to sensory cortex might act to stabilize sensory representations. This could lead to the exact same result without representational stability explicitly constraining behavioral perception. Indeed, I suspect sensory perception constrains behavior much less than simply understanding how to associate the different pulse numbers to arbitrary motor actions, which likely occurs in ALM or other regions outside of vS1.

I still find the results interesting and think they will be of note to the field, but I do think the abstract/discussion should be revised to consider these possibilities and dial back the claims to those that are fully supported by the data.

We thank both reviewers for this comment – this was R2’s first comment as well, and we have altered the text to ensure we do not make causal claims regarding the assorted factors that we find differentiate between learners and non-learners (i.e., inter- and intra-session stability and decoding ability). Changes were made to the title, abstract, introduction, results, and discussion to clarify that learning is associated with representational stability rather than caused by it.

Actual changes:

Title: *Learning in a sensory cortical microstimulation task is associated with elevated representational stability*

Abstract (final sentence, removed causal implication): *Thus, greater stability in the stimulus response is associated with faster learning in a sensory cortical microstimulation task.*

Introduction: *Together, our results show that mice that learn the task exhibit greater stability of evoked activity in sensory cortex.*

Discussion: *We show that over the course of optical microstimulation task training, evoked activity in the stimulated region of vS1 exhibits instability both within and across sessions. We find that greater stability in optogenetically evoked activity is observed among mice that successfully learn to use evoked activity to perform a task. Thus, learning is associated with more stable sensory cortical representations..*

Finally, throughout the results, we tried to ensure that there were no definitive statements about causality.

At the same time, we do think the data do exclude some possibilities regarding the impact of stability. First, even the fastest learning mouse did not attain performance > 75% until day 4 (Figure 1f). Yet the discrepancies in stability among opsin non-expressing neurons both within (Figure 5e) and across (Figure 6f) sessions was present on the very first day. At the same time, as can be seen in Fig. 6c and 6d, which shows average population response correlations across days, there is a measurable increase in stability from the first few days to the last few days among opsin-responsive neurons among learners (there was insufficient imaging time among non-learners to perform comparable analysis).

We feel the most likely scenario is one where both things happen: baseline instability impacts learning, but as animals learn, this can lead to representational stabilization. As mentioned, this could be because well-trained animals exhibit greater motor stereotypy and, presumably, more consistent levels of motivation/engagement. Another interesting possibility would be attentional modulation, which is known to increase similarity between certain cortical responses.

Ideally, one would directly manipulate stability of response (e.g., using two-photon optogenetics to drive consistent and variable patterns of activity), though versions of this experiment we can think of are technically very challenging.

2) I was intrigued by the authors' approach of measuring representational stability by direct cortical stimulation. However, one worry I had about their approach was that exciting large numbers of nearby excitatory neurons could generate localized epileptiform activity which might interact with their

experimental measures. Did the authors test in any way for the presence of hypersynchronous interictal activity?

In 2018/9, we performed a comparison of various transgenics as we were picking a GCaMP transgenic for the lab. A major concern was epileptiform activity that had been observed in some of the candidate lines (Steinmentz et al., 2017, *eNeuro*, Fig. 3). Among the reasons we ultimately picked the Ai162 X Slc17a7 line is because it did not exhibit such activity (Daigle et al., 2018, *Neuron*, Fig. 7A). We observed such activity in our lab in some, but not all, Emx1-Cre X Ai94 mice. Specifically, we observed synchronized surges in $\Delta F/F$ across all ROIs in a field of view.

However, as described, synchronously activating a large number of excitatory neurons could still generate localized epileptiform activity in our transgenic line. To convince ourselves this was not the case, we performed further population-level analyses. Below, we show an example response trace averaged across all ROIs in the field of view. We show an example ~ 45 s of recording (a), and the stimulus-aligned response across all 9-pulse trials (b) for a typical session.

The vertical cyan bars indicate the stimulation epoch, with the number of pulses delivered shown above (the first stimulus in *a* has zero LED pulses, only the shutter and masking flash were activated). Had our photostimulation triggered epileptiform activity, we would expect to see non-stimulus-aligned moments of high $\Delta F/F$ for a single imaging plane with a periodicity of ~ 1 -5s (~ 0.2 -1 Hz; see Steinmentz et al., 2017, *eNeuro*, Fig. 3 and Daigle et al., 2018, *Neuron*, Fig. 7A). This was not the case either in these examples or in others sampled throughout our dataset.

3) In a number of cases, the authors restricted analyses to the top 5th percentile of responsive neurons. It would be helpful to carry out the same analyses for all responsive neurons in each of these cases. Selecting one tail of the distribution can cause iceberg effects and either enhance or obscure observed effects.

We thank the reviewer for this comment as we realized that we were insufficiently clear throughout the manuscript regarding which neurons were used for which analyses; we've checked carefully and hope that it is sufficiently clear now.

In all cases except Figs. 4 and 7, we used all the neurons that were definitely opsin positive or opsin negative (as described in the methods, neurons could be ambiguous with respect to opsin expression due to intermediate levels of co-expressing fluorophore were excluded from all analyses).

The two figures where we did use the top 5% were Figs. 4 and 7.

In the case of Fig 4f, we previously included the bottom 95% and top 5%. This panel now examines all neurons and the top 5% to illustrate that effects are only observed in this smaller population. In Fig. 4g, h, we also previously only included the top 5% of responders. We've now changed this to include all neurons, and adopted the analysis suggested by Reviewer #2 (break up by early and late, as in Fig. 7d, e).

In the case of Fig. 7b, d we used the top 5% decoding scores taken across all neurons. The reasoning was that the rest of the iceberg, as in Fig 4f, did not exhibit substantial differences between learners and non-learners. Indeed, including all neurons brought the mean decoding down to near-chance (0.5), where all but the best-performing decoding neurons are. We therefore feel in this case, restricting the analysis to the top 5% makes sense – this is where the action is. Furthermore, we complement this analysis with one that calculates the fraction of neurons having a score of a certain level (AUC = 0.75; Fig. 7c, e) which replicates the late difference in decoding score among the top 5% of O- neurons, but not the early difference among O+ neurons. Finally, we include Fig. 7a to explicitly show the reader an example population and that there is indeed a small population of neurons that are more interesting than the remainder of the population.

4) The authors used almost exclusively male mice (except for one lone female). NIH policy is to use a sex-balanced sample unless there is a clear scientific justification. Maybe I missed it, but was there any justification for using such a biased sample?

We try to use a sex-balanced sample in most of our studies, but for imaging experiments with water restriction, if we wish to start with young (7-8 week old) animals, females of this strain (which for both sexes are a few grams below C57/black6) will often not make our weight criterion (15g) to start surgeries. Animals that weigh less often have smaller skulls, which complicates window placement, they tend to fare worse post-op, and they have to be temporarily removed from water restriction due to crossing below our weight threshold (20% drop below baseline) more frequently than heavier animals. Alternatively, we could have used older females, but since this is a plasticity study, we wanted to use animals consistent in age. Nevertheless, we appreciate the reviewer's point and want to make it clear that we strive toward sex-balanced samples.

Minor typos/corrections:

Line 515: extra "X" in the the JAX number for SLc17a7-Cre

Line 625: please cite prior art or describe the circuit

Line 639: extra "a" between "from" and "the"

Thank you for catching these; they are all fixed now.

Reviewer #2 (Remarks to the Author):

Pancholi and co-workers investigate chronic remodelling of sensory neural representations in mice that perform a learning paradigm. In this discrimination task, optogenetic stimuli are directly delivered to populations of cortical neurons, serving as cues. This elegant experimental design explores activity dynamics of neuronal populations with concurrent behavioural testing and allows reliable investigations of cortical function in a relevant behaviour. Mice show heterogeneous learning capabilities and are categorized as learners and non-learners. The authors compare representational remodelling in cortical neurons and report as main finding that animals which successfully learned the task show a higher representational stability than mice that do not learn the discrimination task in the time provided. They conclude that “the rate at which sensory cortical representations change constrains learning and impacts behavior”.

The experiments are technically very demanding and of high standard, the analyses appear careful and thorough. The authors are congratulated for this effort. Also, the topic and question are timely and of high general interest. However, while reading the manuscript, I had problems in following the logic and was left unsure if their data really supported their conclusions. As such, I do not think this manuscript, at least in its current form, is suitable for publication in Nature Communications.

Main concerns:

1) The title and abstract claims that there is a causal relationship established between the stability of representations that underlies the ability of a mouse to learn or not. I do not think this study really allows to differentiate if the learning fails due to more unstable representations or the representations fail to stabilize due to the inability to learn. It would be safer to talk about a correlation between these measures and not to suggest any causal relationship.

This was R1’s first comment as well; please see the extensive reply there.

2) The study design, using direct optogenetic activation of the cortex as cues, can offer advantages compared to using sensory stimuli (which are processed then in parallel in multiple brain areas) and in particular to electrical microstimulation. Electrical microstimulation is supposedly rather unstable due to gliosis, as outlined in the introduction.

However, I am not convinced that the optogenetic stimulation (at least as used in this study) is really much better. The design of this study (and other direct stimulation studies) makes the implicit assumption that the set of stimuli that is given to the brain always leads to the same set of activation patterns in the

directly stimulated, “first order” neurons. The nice thing here is that the “input” to the brain is always the same and stable.

We apologize for the lack of clarity, as this was not at all our intent. The input pattern definitely changes, as you point out below, and we sought to characterize this change. We’ve tried to be more explicit about this in the text (see below).

However, I am not sure that this is what the data shows. I have the impression that during the time course of the experiment a substantial adaptation of the “first order” neurons occurs and therefore leads to a constantly changing “input” to the brain, that the mice then are supposed learn and discriminate. Although not directly quantified, I think that Fig. 4e left (although this plot then should be re-sorted for each session), Fig. 4g (although here only top 5% are shown) and most strikingly Fig. 7a top (although here only the decoding power per cell is shown), indicate that the “first order“-activity pattern that is imposed by a given stimulus changes massively throughout the experiment. If I am not completely misinterpreting Fig. 7a, it says that at the end of the experiment, there is hardly any Opsin+ neuron left where the high pulse and the low pulse patterns induce any differential activity. Here, we are talking about the directly stimulated neurons!

We completely agree that the “input” is changing. Specifically, activity in opsin-expressing neurons declines, more so than in opsin non-expressing neurons; showing this decline is the main point of Fig. 4c-f. To make this clear, we now emphasize that both opsin-expressing and opsin non-expressing neurons show reductions in responsiveness during training, and we also added Extended Data Fig. 7, which shows that at all stages of training, activity levels are quite different across pulse counts.

We’ve altered the text to make this clear:

*Restricting our analysis to trials with 9 photostimulus pulses, which were present in all sessions, we identified both **opsin-expressing and opsin non-expressing neurons** with stable, increasing, or decreasing responses to photostimulation (**Fig. 4c-e**). Aggregate responsiveness did not change significantly in either the **opsin-expressing or opsin non-expressing population**, though neurons with the highest responsiveness across sessions (neurons with an overall responsiveness in excess of the top 5% or 1% observed across all sessions; Methods) did show declines in responsiveness (**Fig. 4f**). Despite this decline in responsiveness, high pulse counts evoked more activity among both **opsin-expressing and opsin non-expressing neurons** than low pulse counts at all stages of training (**Extended Data Fig. 7**).*

The reviewer is correct that decoding ability declines, largely due to declining activity among even the directly stimulated neurons. Nevertheless, a small population of highly responsive and efficiently decoding neurons persists (Fig. 7a, top). Even if only 1% of

neurons have an AUC > 0.75, this is still ~50 neurons (assuming 5,000 opsin-expressing neurons). Given the reliability with which small groups of vS1 L2/3 neurons can drive behavior (see Dalgleish et al., 2020, *eLife*), this should still suffice to perform well at the task, as pooling should produce decoding that exceeds decoding by individual neurons. We actually think this is potentially interesting: cortex may have mechanisms for confining overly redundant activity to a smaller set of neurons, leaving the remainder of the population available to encode other inputs.

This point must be clarified and directly addressed. A quantification as shown in 2F (which I guess is from the beginning of the experiment) should be shown for all the time points throughout the experiment. If my suspicion is right, I am afraid that the magenta curve would be essentially flat and low in the last sessions.

To be clear, Fig. 2f (and 2b-2h) shows evoked activity in well-trained mice. This is now clarified in the relevant section of the text:

In well-trained mice performing the final stage of the task (Fig. 1e,f), neurons exhibited diverse responses to photostimulation (Fig. 2b, c).

Furthermore, we added Extended Data Fig. 7 which shows the response to different photostimulus pulse counts over training; even very well-trained animals exhibit clear increases in evoked activity with greater pulse counts.

Also, in Fig. 4g, this adaptation in the top 5% seems to be quite different throughout the initial training and therefore could be a major factor in determining what is a learner and a non-learner. Here, the authors decided to test for differences in the average throughout all sessions not revealing any difference between learners and non-learners. However, a test design as used in 7b-e comparing start and end would likely be significant.

Per R1 point 3, we've now removed the use of the top 5% of responsive neurons for this analysis and have changed to use all neurons (this is consistent with the subsequent learner v. non-learner comparisons in Fig. 5, 6, though in Fig. 7 we do still use top 5% *by decoding ability*; for more details, see R1 point 3 response). With all neurons, we see that responsiveness is similar for opsin-expressing neurons (Fig. 4g). Among opsin non-expressing neurons, activity appears lower among non-learners (Fig. 4h), though it is not significant and the discrepancy is mostly due to a single animal with high responses.

Also, it could be interesting to patch the cells in brain slices to see if the opsin-expressing, stimulated neurons would show strong changes in their intrinsic excitability.

Part of the motivation for including the whisking neuron supplement (Extended Data Fig. 9) was to identify whether cell-intrinsic mechanisms (e.g., changes in excitability) could

be contributing to the response changes we observed over training. If this were the case, opsin-expressing neurons responsive to whisking should show differential whisking responses over training as well. However, we found that the fraction of whisking responsive neurons and their encoding scores were comparable between the opsin-expressing and opsin non-expressing populations and both metrics remained stable over the course of training.

In addition, we've added data on calcium event rates during the intertrial epoch ('spontaneous' activity rates), which shows that these are also comparable among opsin-expressing and opsin non-expressing neurons (Extended Data Fig. 9f, g). The stability of event rates over the course of training suggests that the observed declines in responsiveness are not due to overall changes in excitability specific to opsin-expressing or opsin non-expressing neurons.

Together, this was really the elephant in the room, when reading the manuscript. Also, the interpretation of their findings would probably change a lot, when also considering the stability of the input pattern as a crucial parameter that can impact on learning. It appears plausible that it is more difficult to learn to discriminate a moving target as a stable one.

We failed to make it clear the "input" pattern – activity in the opsin-expressing neurons – is indeed changing throughout training. We hope the reviewer finds this idea is more clearly explained and communicated throughout the text.

3) The third point goes along a similar line: What could be important factors, that differentiate learners from non-learners (apart of the correlation with the stability of activity patterns in non-opsin neurons and eventually faster or slower rundown of imposed activity patterns in "first order" neurons)? Classically, one would consider the similarity or dissimilarity of activity patterns that are elicited by the different cues as a prime parameter how easy it is for the mouse to also discriminate behaviourally. Admittedly, often it is possible to decode the different stimuli from neuronal activity already before training, so that there would be no "need" to change the representations, but it has been shown to nevertheless occur (e.g. Huber, ..., Peron, et al, 2012, Nature; Poort et al., 2015, Neuron) and it would be therefore something obvious to look at. This aspect is touched upon in the very last figure quantifying the single-cell decoding ability. But in most of the manuscript's figures only the activity patterns evoked by the 9 pulse stimuli are shown and quantified.

I think it is important to provide also more detail on the structure of the activity patterns evoked by the other stimuli. Perhaps it is the stability of activity pattern evoked by the 1 pulse stimulus, that better differentiates learners from non-learners? Or it is the structure of the various activity patterns, and not the overall

count of elicited activity. I would not be surprised, that some sort of normalization of overall activity is happening during the course of the experiment and the differentiation in the last sessions is based on the similarity or dissimilarity of the patterns.

We apologize for the lack of clarity. The activity patterns evoked by the 9-pulse stimuli are used in most figures when comparing learners to non-learners because non-learners were never progressed to pulse counts beyond the 9/0 detection task. We have tried to clarify this point. In the text accompanying Fig. 1, we now state:

Mice that failed to improve on this detection task were classified as 'non-learners' (n = 5). These mice were only exposed to the 9-pulse vs. 0-pulse stage of the task and were removed from the training cohort within 1-2 weeks.

Furthermore, we've added a phrase in each relevant results section stating that that section uses data from only well-trained mice, which we agree was unclear in the original version.

In addition, we have added Extended Data Fig. 7 to provide more detail on the structure of the activity patterns evoked by the other stimuli for animals that learned the task. As can be seen, there is a decline in overall responsiveness across the board, though responses do remain substantial and distinguishable.

Thus, we hope it is clear that it is not possible to perform the analyses the reviewer proposes, since none of the non-learners progressed beyond the 9-pulse/0-pulse stage. This is also the reason we exclusively analyze 9-pulse trials when comparing various types of animals – this is the one trial type present in every experimental session.

Minor points:

The binary categorization of mice is somewhat arbitrary. Usually, most mice learn if you give them enough time. A non-binary, parametric correlation of learning (at least during the first 8 days) and the various neurometrics can be considered.

Per the reviewer's request, we performed analyses correlating performance on the final three common days across all mice (days 6-8) with the various metrics of neural activity / similarity / decoding. Unfortunately, this analysis did not yield significant correlations in any of the cases. This seems to be because at that particular timepoint, the mice that do learn are actually not differentiable; if we use a later window (e.g., 10-12), we see significant correlations but have to drop several animals.

As can be seen below, we examined the various parameters that we showed in the main figures. We provide the Pearson correlation and the P-value for this measure.

Compare to Fig. 4g, h

Compare to Fig. 5e

Compare to Fig. 6f

Compare to Fig. 7d,e

To better understand the learning process, it would be interesting to show not only overall performance, but also the hits, misses, FA, CR for the two categories of stimuli. I could imagine that there is some asymmetry in learning: The 9 pulse stimuli are learned initially and when introducing the 1 pulse stimulus the overall performance is mainly due to reduce the rate of FA.

We've added Extended Data Fig. 5, which shows the "hit" rate (the rate of correct, rightward licks on high pulse count trials) and "false alarm" rate (the rate of incorrect, rightward licks on low pulse count trials). The "false alarm" rate is relatively flat following transition from 9/0 to 9/1, but the "hit rate" drops in all but one mouse. Interestingly, this suggests that the mouse make errors more frequently in high, and not low intensity stimulation trials.

This is corroborated by the psychometric curve shown in in Fig. 1g, where the tendency to lick right (report high intensity stimulus) is ~85% in well-trained mice performing the final stage of the task for 9 pulses, and ~10% for 1 pulse trials. Thus, mice make more mistakes (15%) on 9 pulse trials than 1 pulse trials.

I am surprised that many mice still perform the task above chance after the lesion. When removing all opsin expressing neurons, the mice should have no way to discriminate the trials. Was the lesion incomplete or what is the authors' interpretation?

We imaged histological sections in all but one of the lesioned mice; as seen in the newly added Extended Data Fig. 6, some lesions were not complete, so that some opsin-expressing neurons were spared.

For the cited literature: Representational drift also has been recently shown in auditory cortex (Aschauer et al., 2022, Cell Reports), and earlier as in the cited studies in visual cortex (Montijn et al., 2015 Cell Reports; albeit not that explicitly named).

Thank you for the additional references, they have been added.

In Fig. 1E, I would put the scheme how the stimuli are introduced on top and somehow graphically fuse it better to the learning curve. For a long time, I associated the scheme with panel F. Also, one could use the same shapes in the scheme and in F.

Figure 1e and 1f have been adjusted to incorporate this suggestion.

In Fig. 3C there is an average trend towards more activity in right lick trials. Why is that? A matter of the hemisphere or of the contingency? I guess the interpretation is that in those trials where more activity as average is elicited, this may be easier confused with a right lick trial. But the actual discrimination of the stimuli may not depend on the overall rate of activity, despite the fact that higher activity rates are correlated with higher pulse numbers.

This was the question this figure aimed to ask: does the amount of evoked activity bias behavior? Per this analysis, for any given pulse count in well-trained animals (i.e., in the final stage of the task), trials where the animal licked right showed higher levels of evoked activity than trials where the animal licked left. This observation suggests that animals are using the activity evoked by photostimulation in cortex to perform the task.

In Fig. 5D, it is not clear what is plotted at the bottom of the correlation matrices. The average of the columns? What would that tell us? How representative a given trial is compared to all others? Perhaps expand the legend a bit more. Also, what are the trials with very low correlations in the matrices?

The legend has been updated to clarify what this is; hopefully this language is clearer:

bottom, average correlation with all other trials over the course of the session

We include this panel to show how correlation fluctuates over the course of the session. For instance, there is a period for the first ~70% of the session where the opsin non-expressing neurons show higher pairwise correlations than the opsin-expressing neurons.

We do occasionally see trials with very low responses; this may be due to the interaction of up/down states with our photostimulus (see Mateo et al., 2011 for evidence of this). These trials drive the low correlations.

Fig. 6, this analysis considers the response patterns to the 9 pulse stimulus? Please indicate clearly in legend.

Thank you for pointing this out; we explicitly stated this in describing Fig. 5 so it was indeed a glaring omission for Fig. 6.

The figure legend has been altered:

*Pearson correlation between mean evoked $\Delta F/F$ response on day 1 and example subsequent days **for 9-pulse trials**.*

The accompanying text has also been updated:

*To obtain an overall measure of stability for a given session, we computed the mean trial-to-trial correlation of response vectors for a given session in all mice, **restricting our analysis to 9-pulse trials**.*

REVIEWER COMMENTS

Reviewer #1 (Remarks to the Author):

The authors addressed the majority of the reviewer comments thoroughly.

One remaining statistical concern: from point #3 in my 1st round review, I still don't like the use of arbitrary cutoffs (e.g., top 5%, $AUC > 0.75$) rather than all significantly responding/coding cells. If they want to stick with cutoffs, they should try a reasonable range of cutoff values and show that the results are robust (this helps avoid accidental p-hacking).

Otherwise, this is an excellent manuscript and I look forward to seeing in print.

Reviewer #2 (Remarks to the Author):

Panchioli and co-workers submitted a revised version of their manuscript. Many of my concerns were sufficiently addressed, however, I feel that still several points were left open.

Again, I think this is an interesting study and the data will be of interest. However, a successful paper will not only have 'made it' into a particular journal, but is also well appreciated by the readers. My comments are primarily aimed to help the authors to reach this goal.

Main concerns from first review:

1) Relationship between learning and representational stability is correlational.

Adequately addressed.

2) The main effect in their data is a massive adaptation to optogenetic stimulation that occurs to a similar level in 'learners' and 'non-learners' and that affects even directly stimulated, opsin-expressing neurons. This major effect in their data is almost hidden in the way the manuscript is presented.

The adaptation effect is surprising to me and to the best of my knowledge also novel to the field. It is obviously the authors' free choice to set their focus on whatever aspect in their data, although my impression is that the reported differences in learners and non-learners are comparably minuscule. But I urge the authors to at least clearly acknowledge this major effect, even if they think it is of less interest. In the revised manuscript the authors made small changes to address this point, however, I think the overall presentation is still misleading.

E.g. in the introduction they write: "However, electrical microstimulation of cortex evokes activity among a sparse and spatially distributed population²⁷, suffers from instability due to slowly emerging gliosis²⁸, and produces a prominent stimulus artifact that complicates simultaneous recording²⁹. Optical microstimulation of cortex^{25,26,30,31} overcomes these constraints and...."

This sentence, at least in my eyes, implicitly states that the method of microstimulation was used in order to evoke stable patterns of activity in the cortex. In the rebuttal, the authors state that: "The input pattern definitely changes, as you point out below, and we sought to characterize this change." If the change in optogenetically evoked activity was indeed at the focus of the study, I would suggest to point this out in the introduction, in order to facilitate the understanding for readers.

Moreover, I would really suggest to address this point in the corresponding results section and/or discussion section, by including an explicit statement that this massive decline in optogenetic responses, which is seen in learners and non-learners during the time course of the experiment is either a) well known and has been demonstrated before and therefore to be expected (bolstered by several citations), or, b) is indeed a novel finding, but the focus of the analysis is on the differences between learners and non-learners for this and that reason. I am convinced that this would help many readers to be able to better follow the reasoning in the manuscript.

The, in my opinion overly strong, focus on differences between learners and non-learners is also still reflected in the title of Fig. 4: "Figure 4. Population dynamics over the course of training and their impact on learning.", where actually no significant differences between learners and non-learners are reported. Furthermore, it remains even unclear if this massive decline in optogenetically induced activity could be even observed in mice that are not involved in any explicit training.

3) Many analyses are only on 9-pulse patterns.

Adequately addressed and/or clarified.

Minor points:

A non-binary, parametric correlation of learning (at least during the first 8 days) and the various neurometrics can be considered.

As requested, the authors did the correlation analysis between performance and the various parameters describing the population activity.

However, none of the analyses showed a convincing and statistically significant correlation. For example the correlation: learning vs inter-day stability of representations, which is at the core of the claims.

This is obviously unsettling.

One reason is that it is unclear how the data was analyzed: How can it be that red dots (learners) and black dots (non-learners) do not fully separate along the y-axis (Performance on day 8), when the result section says: line 141: "Mice were classified as 'learners' (n = 6) if their performance improved over the first week of training. Mice that failed to improve on this detection task were classified as 'non-learners' (n = 5)." ???

The other reason is that the reported correlation between learning and stability of representations is apparently (??) done across different time points. The analyses of neural activity patterns is reported only during early phases of training (≤ 8 days, eg. Fig. 5, 6, 7), but the final discrimination between learners and non-learners was done then a few days later? Why then not also show the activity-data from non-learners for later time points? At least the results should be stated in a way that makes this clear: "Differences in activity patterns during the first week (as shown) predict learning rates at later time points".

The current presentation of the data, analysis and in particular the stratification of mice in learners and non-learners is in my opinion not clear and confusing. This must be addressed and clarified.

In Fig. 3C there is an average trend towards more activity in right lick trials. Why is that? A matter of the hemisphere or of the contingency?

I understand that the authors wanted to test if higher activity levels are correlated with right licks (as reinforced), which is consistent with an interpretation that the evoked activity is reflecting sensory information used by the animal to make its decisions. However, even in sensory cortices, 'lick-neurons' have been reported and it is therefore conceivable that activation of one hemisphere could bias movements to a particular side. My question was intended to motivate the authors, if available, to provide further evidence for their interpretation, such that similar observations are made, even if the contingency of the stimuli is switched, the direction of the licks was changed or anything alike.

In Fig. 5D, it is not clear what is plotted at the bottom of the correlation matrices.

Unclear if the legend is now more informative. I assume it is indeed the average of the columns (or rows) of the correlation matrix? Also still unclear what that metric should show. Eg. Consider a matrix with correlations smoothly falling off the diagonal. This metric would be constant over time, despite actual "drift".

Also, "representational drift" is usually used to describe the reconfiguration of population activity over sessions and days. Analyzing the variability in neuronal responses within several seconds to minutes, a term like "Trial-to-trial variability of population responses" would be probably more familiar to most readers.

All other minor points were adequately addressed.

REVIEWER COMMENTS

Reviewer #1 (Remarks to the Author):

The authors addressed the majority of the reviewer comments thoroughly.

One remaining statistical concern: from point #3 in my 1st round review, I still don't like the use of arbitrary cutoffs (e.g., top 5%, AUC > 0.75) rather than all significantly responding/coding cells. If they want to stick with cutoffs, they should try a reasonable range of cutoff values and show that the results are robust (this helps avoid accidental p-hacking).

We use arbitrary cutoffs in three places. In all cases, this is because the effect of interest is confined to neurons that are either very responsive or very discriminative, and so we arbitrarily partition the data to emphasize high responders/good discriminators.

First, we use an arbitrary cutoff in responsiveness (must have response probability > 0.25) to restrict our analysis of discriminability between right and left licks (**Fig. 3**) to responsive neurons. Neurons that respond less often do not discriminate between left and right licks as effectively, and as we include more of them, the effect weakens. On the other hand, as we use a more stringent criteria and admit only very responsive neurons, the effect becomes larger. We include below the population data (legend as in **Fig. 3d, e**) for the case where all neurons are included and the one where the cutoff was more stringent (response probability > 0.5). As can be seen, the conclusion that opsin non-expressing neurons show a difference whereas opsin expressing neurons do is robust, and still holds even when all neurons are included (left case).

We also use an arbitrary cutoff in the analyses looking at changes in responsiveness (**Fig. 4**). Here, we include a sorted version of the entire population in an exemplar animal (**Fig. 4e**). Our goal in providing this figure was to illustrate to the reader that

there are populations with distinct dynamics over the course of training. By averaging over the entire population (**Fig. 4f**, left), we see that the distinct dynamics wash out and there appears to be minimal change in responsiveness across training. **Fig. 4e** should make clear that the largest changes are in the most responsive neurons, and that an analysis restricted to these neurons is appropriate, which is why we then use cutoffs of 5% and 1% (**Fig. 4f**, middle and right). We show both because we agree with the reviewer's point and felt that including both would make clear that the higher the cutoff, the more pronounced the effect, but that the reported effect would be arbitrarily dependent on the exact cutoff chosen.

The last place we use an arbitrary cutoff is **Fig. 7b, d**. We use the 5% most discriminative neurons because the majority of the effect is confined to the most discriminative cells, which we again show for all cells in an example animal (**Fig. 7a**). If we use a more stringent (e.g., 1%) threshold, the example animal shows that the effect will become more pronounced, whereas a less stringent threshold (e.g., 10%) will dampen the effect. The logic is analogous to **Fig. 4** (above) – we show the raw data in an example animal (a) sorted by AUC to illustrate that at least for the top neurons (which we show in the zoom), there is indeed a change. **Fig. 7c, e** analyze the data in a complimentary manner – using a cutoff AUC value instead of a percentile of the distribution.

Otherwise, this is an excellent manuscript and I look forward to seeing in print.

Reviewer #2 (Remarks to the Author):

Panchioli and co-workers submitted a revised version of their manuscript. Many of my concerns were sufficiently addressed, however, I feel that still several points were left open.

Again, I think this is an interesting study and the data will be of interest. However, a successful paper will not only have 'made it' into a particular journal, but is also well appreciated by the readers. My comments are primarily aimed to help the authors to reach this goal.

Main concerns from first review:

1) Relationship between learning and representational stability is correlational.

Adequately addressed.

2) The main effect in their data is a massive adaptation to optogenetic stimulation that occurs to a similar level in 'learners' and 'non-learners' and that affects even directly stimulated, opsin-expressing neurons. This major effect in their data is almost hidden in the way the manuscript is presented.

The adaptation effect is surprising to me and to the best of my knowledge also novel to the field. It is obviously the authors' free choice to set their focus on whatever aspect in their data, although my impression is that the reported differences in learners and non-learners are comparably minuscule. But I urge the authors to at least clearly acknowledge this major effect, even if they think it is of less interest. In the revised manuscript the authors made small changes to address this point, however, I think the overall presentation is still misleading.

E.g. in the introduction they write: "However, electrical microstimulation of cortex evokes activity among a sparse and spatially distributed population²⁷, suffers from instability due to slowly emerging gliosis²⁸, and produces a prominent stimulus artifact that complicates simultaneous recording²⁹. Optical microstimulation of cortex^{25,26,30,31} overcomes these constraints and....."

This sentence, at least in my eyes, implicitly states that the method of microstimulation was used in order to evoke stable patterns of activity in the cortex. In the rebuttal, the authors state that: "The input pattern definitely changes, as you point out below, and we sought to characterize this change." If the change in optogenetically evoked activity was indeed at the focus of the

study, I would suggest to point this out in the introduction, in order to facilitate the understanding for readers.

We've increased the emphasis on the change in responsiveness (the 'adaptation effect' mentioned by the reviewer) both the abstract and introduction. We also thank the reviewer for pointing out an area of potential confusion. We did not mean that optical microstimulation results in stable activity. Rather, we only meant to point out that electrical microstimulation approaches have unstable stimulus input over time because of mechanisms like gliosis. Therefore, with our approach, any observed changes in evoked activity should be more readily attributable to intrinsic changes in the studied circuit than with past approaches. We've altered the language in this portion of the introduction to make this point clearer:

However, electrical microstimulation of cortex evokes activity among a sparse and spatially distributed population²⁷, suffers from instability due to slowly emerging gliosis²⁸, and produces a prominent stimulus artifact that complicates simultaneous recording²⁹. Optical microstimulation of cortex^{25,26,30,31} overcomes these constraints: opsin expression can be stable for the duration of the experiment; viral opsin delivery allows for refined spatial and genetic control of evoked activity; optical microstimulation is compatible with chronic concurrent recording via large-scale calcium imaging³².

Moreover, I would really suggest to address this point in the corresponding results section and/or discussion section, by including an explicit statement that this massive decline in optogenetic responses, which is seen in learners and non-learners during the time course of the experiment is either a) well known and has been demonstrated before and therefore to be expected (bolstered by several citations), or, b) is indeed a novel finding, but the focus of the analysis is on the differences between learners and non-learners for this and that reason. I am convinced that this would help many readers to be able to better follow the reasoning in the manuscript.

We agree this is an important point and have emphasized it now in several ways. In the abstract (emphasizes the change rather than learner/non-learner):

Population activity levels across training declined rapidly, with the most active neurons showing the largest declines.

It is mentioned in the introduction's summary paragraph:

Photostimulus-evoked activity declines rapidly over the course of learning, especially among the most responsive neurons.

We also have a paragraph in the discussion dedicated to this sparsification, which we've elaborated now and added references showing that in past studies, both increases and declines in activity have been observed:

*We observe a large degree of sparsification in the photoresponsive population over the course of the first week of training, both in terms of the size of the responsive population and the typical response for a given neuron. With novel natural stimuli, both increases and decreases in responsive populations have been observed. For example, novel, behaviorally-relevant tones become overrepresented across learning in auditory cortex,^{59,60} whereas whisker representations can shrink following environmental enrichment in vS1⁶¹. Direct stimulation of cortex or of cultured cortical neurons can drive both increases⁶²⁻⁶⁴ and reductions^{65,66} in responsiveness. Candidate mechanisms for declining responsiveness include increased inhibitory input onto pyramidal neurons⁶⁷, reduced excitability^{68,69}, and reductions in excitatory connectivity⁶⁵. Reduced excitability is a less likely mechanism in our experiment, given the lack of change in whisking responses and stability of spontaneous activity (**Extended Data Fig. 9**), which should both be impacted if global excitability changed. Given that the most active neurons experience disproportionately larger response declines, sparsification is unlikely to be due to global mechanisms, such as a general increase in inhibition. Instead, cell-specific changes likely account for the observed dynamics, such as targeted changes in inhibition.*

The, in my opinion overly strong, focus on differences between learners and non-learners is also still reflected in the title of Fig. 4: “Figure 4. Population dynamics over the course of training and their impact on learning.”, where actually no significant differences between learners and non-learners are reported.

We have now de-emphasized the learner/non-learner contrast until Fig. 4g, h, removing “impact on learning” from the Fig. 4 title since, as the reviewer points out, the results show no significant difference between the two.

Furthermore, it remains even unclear if this massive decline in optogenetically induced activity could be even observed in mice that are not involved in any explicit training.

While this was not the focus of the present study, we have a second manuscript under review where we photostimulated an identically transfected/implanted mouse using 9 pulses with no explicit training (Pancholi et al., 2022, *bioRxiv* preprint at: <https://www.biorxiv.org/content/10.1101/2022.11.09.515803v1>). There, we saw no change in responsiveness in animals with no explicit training, suggesting that the decline in **Fig. 4** may be dependent on task-engagement as the reviewer suggests. We are planning to explore this in more detail in a future project.

3) Many analyses are only on 9-pulse patterns.

Adequately addressed and/or clarified.

Minor points:

A non-binary, parametric correlation of learning (at least during the first 8 days) and the various neurometrics can be considered.

As requested, the authors did the correlation analysis between performance and the various parameters describing the population activity.

However, none of the analyses showed a convincing and statistically significant correlation. For example the correlation: learning vs inter-day stability of representations, which is at the core of the claims.

This is obviously unsettling.

We agree, and this is an unfortunate weakness of the dataset. In retrospect, if we had imaged the non-learners for, say, 15 days, that would have been the ideal day to compare across animals. Unfortunately, at day 8, several of the learners were just starting their improvement, so that their performance measured over the entire session was low and not very distinguishable from non-learners. Because our criteria for learners (described in far more detail below) effectively only requires a burst of good performance, overall performance (**Fig. 1e**) which was used in the correlation computations in the previous reply was low in several of the learners, lessening the distinction between the groups.

One reason is that it is unclear how the data was analyzed: How can it be that red dots (learners) and black dots (non-learners) do not fully separate along the y-axis (Performance on day 8), when the result section says: line 141: “Mice were classified as ‘learners’ (n = 6) if their performance improved over the first week of training. Mice that failed to improve on this detection task were classified as ‘non-learners’ (n = 5).“ ???

We apologize for the imprecise language and agree that the distinction between learners and non-learners needs to be very clearly and explicitly explained. When we look at behavior, we examine d-prime for 61 trial windows centered on all trials from trial 31 onto trial 31 before the end of the session. We use this window to both compute a ‘local’ percent correct (which we report as it is the more easy-to-understand metric; **Fig. 1e**, dots) and d-prime. Thus, for any session, we can compute a maximal local d-prime (and percent correct), which lets us capture moments where an animal does really well despite a not-so-great session, a common occurrence in early learning. If the animal exceeds a d-prime of 1.5 over the 61 trial window for 2 days in a row prior to the 8th day of training and continues to perform at this level, we consider this animal a ‘learner’. All such animals continue to improve and become expert at all stages of the task. Animals that fail these criteria are identified as non-learners.

We’ve made several changes to ensure we describe these criteria clearly to the reader.

First, we’ve now updated it in the aforementioned section of the results:

Mice were classified as ‘learners’ (n = 6) if their performance reached criteria over the first 1-2 weeks of training, defined as having at least two consecutive days with a peak d-prime of 1.5 in a 61 trial window (Methods). Mice that failed to improve on this detection task were classified as ‘non-learners’ (n = 5).

In the methods, we now state:

Mice were evaluated using a sliding 61-trial window over which both percent correct (Fig. 1e) and d-prime were calculated. For each session, a peak d-prime across these sliding windows was computed. Mice that had attained at least two sessions with a peak d-prime of 1.5 and continued to have peak d-prime > 1.5 by the 8th session were considered learners; mice that had not yet attained such performance, or had regressed despite two consecutive days of meeting criteria, were considered non-learners.

The other reason is that the reported correlation between learning and stability of representations is apparently (??) done across different time points. The analyses of neural activity patterns is reported only during early phases of training (<=8 days, eg. Fig. 5, 6, 7), but the final discrimination between learners and non-learners was done then a few days later? Why then not also show the activity-data from non-learners for later time points? At least the results should be stated in a way that makes this clear: “Differences in activity patterns during the first week (as shown) predict learning rates at later time points”.

We have ensured that every results section clearly explains what days are used when comparing learners and non-learners.

For **Fig. 4g, h**, we look at days 1-3 and days 6-8. We used two timepoints as the responsiveness is changing so averaging over all common days (1-8) would not be appropriate. We show the full trends for both groups so that the reader can see that the values are instable.

For intraday stability in **Fig. 5e**, we use days 1-8 since the mean trial-to-trial correlation within a session is not changing much over those days.

For interday stability in **Fig. 6f**, we use days 1-3 only because the metric measures how many days until correlation drops below 0.5, so it in fact goes further out (day 8) and uses all the data we have.

For decoding in **Fig. 7d,e** we use the approach we used in **Fig. 4** (compare 1-3 and 6-8 separately) because, again, the values are non-stationary across days. We show these in **Fig. 7b,c** so the reader can see the instability.

In all cases this is stated explicitly in the relevant section of the results.

The current presentation of the data, analysis and in particular the stratification of

mice in learners and non-learners is in my opinion not clear and confusing. This must be addressed and clarified.

We hope we've improved clarity sufficiently with the present revision. We've ensured the date range used is explained in all relevant results sections, and we've more explicitly stated at key transition points that we are looking for differences in early training that predict learning.

In Fig. 3C there is an average trend towards more activity in right lick trials. Why is that? A matter of the hemisphere or of the contingency?

I understand that the authors wanted to test if higher activity levels are correlated with right licks (as reinforced), which is consistent with an interpretation that the evoked activity is reflecting sensory information used by the animal to make its decisions. However, even in sensory cortices, 'lick-neurons' have been reported and it is therefore conceivable that activation of one hemisphere could bias movements to a particular side. My question was intended to motivate the authors, if available, to provide further evidence for their interpretation, such that similar observations are made, even if the contingency of the stimuli is switched, the direction of the licks was changed or anything alike.

We include a delay period between touch and licking precisely to ensure that lick responses are not being included in responsiveness measures. For responsiveness, we only consider the two frames after shutter opening, which precedes lickport availability by ~200 ms (1-2 imaging frames). We monitor licking with a video camera, and all mice we use stop licking prematurely (i.e. prior to the response epoch, Methods) very early in training.

We now clarify this in the legend for Figure 3:

c. Mean photostimulation-evoked $\Delta F/F$ across all neurons during an example session, restricted to timepoints prior to first lick. Box plots as in b. Top, opsin-expressing neurons; bottom, opsin non-expressing.

We've also clarified this at the end of the "Quantifying responsiveness" methods section:

In cases where evoked response levels are reported, only the two frames immediately after shutter opening are used, so that reported evoked activity precedes any licking.

In Fig. 5D, it is not clear what is plotted at the bottom of the correlation matrices.

Unclear if the legend is now more informative. I assume it is indeed the average of the columns (or rows) of the correlation matrix? Also still unclear what that metric should show. Eg. Consider a matrix with correlations smoothly falling off

the diagonal. This metric would be constant over time, despite actual “drift”.

We agree that the utility and clarity of this panel are minimal. We’ve removed it.

Also, “representational drift” is usually used to describe the reconfiguration of population activity over sessions and days. Analyzing the variability in neuronal responses within several seconds to minutes, a term like “Trial-to-trial variability of population responses” would be probably more familiar to most readers.

We’ve removed the term representational drift from all but the first introductory paragraph.

All other minor points were adequately addressed.

REVIEWERS' COMMENTS

Reviewer #2 (Remarks to the Author):

Panchioli and co-workers submitted a second revision of their manuscript. All of my points of concern were now adequately addressed.

I would like to congratulate the authors. A very demanding, but highly interesting study, which will be of interest to a large group of readers.

Panchioli et al:

""""While this was not the focus of the present study, we have a second manuscript under review where we photostimulated an identically transfected/implanted mouse using 9 pulses with no explicit training (Pancholi et al., 2022, bioRxiv preprint at: <https://www.biorxiv.org/content/10.1101/2022.11.09.515803v1>). There, we saw no change in responsiveness in animals with no explicit training, suggesting that the decline in Fig. 4 may be dependent on task-engagement as the reviewer suggests. We are planning to explore this in more detail in a future project.""""

Well this is great. To me it sounds as if this differential effect in adaptation/sparsification between learners (independent if fast or slow) and non-learners (just passively exposed, although they still may undergo some form of non-reinforced learning) may be really worth following up. Interesting!

REVIEWER COMMENTS

Reviewer #2 (Remarks to the Author):

Panchioli and co-workers submitted a second revision of their manuscript. All of my points of concern were now adequately addressed.

I would like to congratulate the authors. A very demanding, but highly interesting study, which will be of interest to a large group of readers.

Panchioli et al:

""""While this was not the focus of the present study, we have a second manuscript under review where we photostimulated an identically transfected/implanted mouse using 9 pulses with no explicit training (Pancholi et al., 2022, bioRxiv preprint at: <https://www.biorxiv.org/content/10.1101/2022.11.09.515803v1>). There, we saw no change in responsiveness in animals with no explicit training, suggesting that the decline in Fig. 4 may be dependent on task-engagement as the reviewer suggests. We are planning to explore this in more detail in a future project.""""

Well this is great. To me it sounds as if this differential effect in adaptation/sparsification between learners (independent if fast or slow) and non-learners (just passively exposed, although they still may undergo some form of non-reinforced learning) may be really worth following up. Interesting!

We thank the reviewer for their help; we feel the final manuscript is substantially improved from this review process. Given the lack of additional requests, we only be changing the manuscript to remove any small mistakes and to comply with word limits, editorial policy, etc.